# Keep It on a Leash: Controllable Pseudo-label Generation Towards Realistic Long-Tailed Semi-Supervised Learning

**Yaxin Hou**[1], **Bo Han**[1], **Yuheng Jia**[1,2,3,*] **Hui Liu**[3], **Junhui Hou**[4]

[1]School of Computer Science and Engineering, Southeast University, Nanjing 210096, China
[2]Key Laboratory of New Generation Artificial Intelligence Technology and Its
Interdisciplinary Applications (Southeast University), Ministry of Education, China
[3]School of Computing Information Sciences, Saint Francis University, Hong Kong, China
[4]Department of Computer Science, City University of Hong Kong, Hong Kong, China
`yaxin@seu.edu.cn, hanbo@seu.edu.cn, yhjia@seu.edu.cn,`
`h2liu@sfu.edu.hk, jh.hou@cityu.edu.hk`

## Abstract

Current long-tailed semi-supervised learning methods assume that labeled data exhibit a long-tailed distribution, and unlabeled data adhere to a typical predefined distribution (i.e., long-tailed, uniform, or inverse long-tailed). However, the distribution of the unlabeled data is generally unknown and may follow an arbitrary distribution. To tackle this challenge, we propose a Controllable Pseudo-label Generation (CPG) framework, expanding the labeled dataset with the progressively identified reliable pseudo-labels from the unlabeled dataset and training the model on the updated labeled dataset with a known distribution, making it unaffected by the unlabeled data distribution. Specifically, CPG operates through a controllable self-reinforcing optimization cycle: (i) at each training step, our dynamic controllable filtering mechanism selectively incorporates reliable pseudo-labels from the unlabeled dataset into the labeled dataset, ensuring that the updated labeled dataset follows a known distribution; (ii) we then construct a Bayes-optimal classifier using logit adjustment based on the updated labeled data distribution; (iii) this improved classifier subsequently helps identify more reliable pseudo-labels in the next training step. We further theoretically prove that this optimization cycle can significantly reduce the generalization error under some conditions. Additionally, we propose a class-aware adaptive augmentation module to further improve the representation of minority classes, and an auxiliary branch to maximize data utilization by leveraging all labeled and unlabeled samples. Comprehensive evaluations on various commonly used benchmark datasets show that CPG achieves consistent improvements, surpassing state-of-the-art methods by up to **15.97%** in accuracy. The code is available at `https://github.com/yaxinhou/CPG`.

## 1 Introduction

Over the past decade, semi-supervised learning (SSL) has emerged as a mainstream paradigm for enhancing the performance of deep neural networks (DNNs) in label-scarce domains like medical diagnosis [1, 2, 3] by leveraging abundant unlabeled data. The dominant SSL framework utilizes unlabeled samples by generating pseudo-labels from high-confidence model predictions [4, 5, 6]. However, conventional SSL methods [7, 8, 9] rely on the unrealistic assumption of balanced and aligned distri-

---

[*]Corresponding author.

39th Conference on Neural Information Processing Systems (NeurIPS 2025).

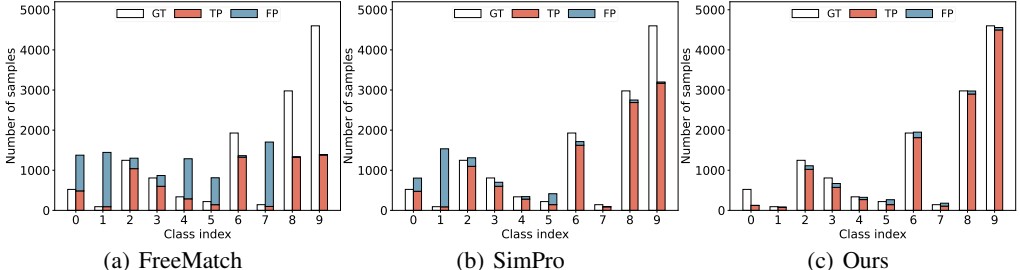

Figure 1: Comparison of pseudo-label predictions among FreeMatch [8], SimPro [13], and our CPG under arbitrary unlabeled data distribution. GT denotes the ground-truth unlabeled data distribution. TP (FP) denotes the predicted true (false) positive pseudo-labels. The dataset is CIFAR-10-LT with $(N_{max}, M_{max}, \gamma_l, \gamma_u) = (400, 4600, 50, 50)$, where $N_{max}$ ($M_{max}$) denotes the number of samples in the most frequent class of the labeled (unlabeled) dataset, while $\gamma_l$ ($\gamma_u$) denotes the imbalance ratio of the labeled (unlabeled) dataset. Our CPG can generate more reliable pseudo-labels than FreeMatch and SimPro in both minority classes like class 1, 2, and majority classes like class 8, 9.

butions for labeled and unlabeled data. Long-tailed semi-supervised learning (LTSSL) [10, 11, 12] relaxes the balance assumption, however, it still suffers from potential distribution mismatches between labeled and unlabeled data.

**A motivating example.** In urban traffic monitoring systems, vehicle images captured by surveillance cameras are classified into distinct categories (e.g., bicycles, trucks, motorcycles) based on their visual characteristics. This presents a long-tailed semi-supervised learning scenario, where labeled data exhibit a long-tailed distribution (with common vehicles like cars dominating and specialized vehicles like ambulances being underrepresented). Meanwhile, unlabeled data collected across different urban environments share the same vehicle categories but may exhibit unknown distributional shifts. These distributional shifts arise from spatial variations (e.g., higher proportions of personal vehicles in residential areas) or temporal fluctuations (e.g., increased taxi frequency during rush hours).

Recent advances in realistic long-tailed semi-supervised learning (ReaLTSSL) have attempted to address the distribution mismatch between labeled and unlabeled data. For instance, ACR [14] adjusts logits based on the distance between the estimated unlabeled data distribution and the predefined anchor distributions, while CPE [15] employs multiple classifiers (experts) to model diverse unlabeled data distributions. Although effective, both methods assume that unlabeled data adhere to a typical predefined distribution (i.e., long-tailed, uniform, or inverse long-tailed), which may not hold in practice. The recent method SimPro [13] alleviates this limitation through an Expectation-Maximization (EM)-based pseudo-labeling method. However, as illustrated in Fig. 1(b), SimPro's distribution estimation ability degrades when unlabeled data follow an unknown arbitrary distribution. In summary, existing methods rely on a confidence threshold to select high-confidence pseudo-labels for distribution estimation, which subsequently guides pseudo-label generation. However, as shown in Figs. 1 and 3, this strategy becomes unreliable when handling the unknown arbitrary unlabeled data distribution, as high-confidence pseudo-labels may still contain substantial errors. These inaccuracies propagate during training, resulting in confirmation bias and model performance degradation.

$\mathcal{D}_u$: Unlabeled dataset   $\mathcal{D}_l(t)$: Updated labeled dataset   $f_t$: Constructed Bayes-optimal classifier   ▨ Identified reliable pseudo-labels

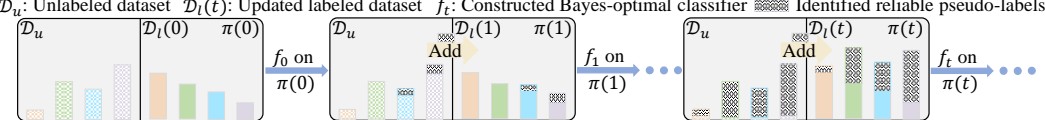

Figure 2: Overview of the controllable self-reinforcing optimization cycle.

To address these challenges, we introduce a Controllable Pseudo-label Generation (CPG) framework that employs a controllable self-reinforcing optimization cycle to handle the unknown arbitrary unlabeled data distribution problem in ReaLTSSL. As illustrated in Fig. 2, CPG utilizes a dynamic controllable filtering mechanism to identify reliable pseudo-labels. These reliable pseudo-labels are combined with the labeled dataset to construct an updated labeled dataset $\mathcal{D}_l(t)$ with a known distribution $\pi(t)$. We then construct a Bayes-optimal classifier $f_t$ via logit adjustment [16] based on the known distribution $\pi(t)$. Note that the Bayes-optimal classifier is unaffected by the unlabeled data distribution. The Bayes-optimal classifier $f_t$, in turn, helps identify more reliable pseudo-labels in the next training step. Through this optimization cycle, an increasing number of unlabeled samples are progressively incorporated as reliable ones, enabling the framework to capture the unlabeled data

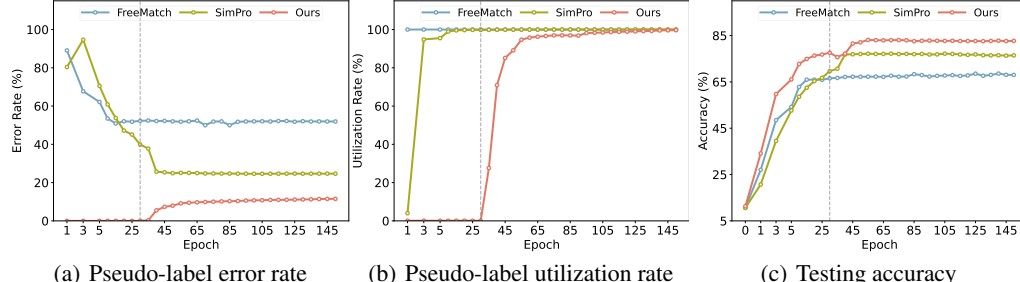

| (a) Pseudo-label error rate | (b) Pseudo-label utilization rate | (c) Testing accuracy |

Figure 3: Comparison of pseudo-label error rate (a), pseudo-label utilization rate (b), and testing accuracy (c) among FreeMatch [8], SimPro [13], and our CPG under arbitrary unlabeled data distribution. The dataset is CIFAR-10-LT with $(N_{max}, M_{max}, \gamma_l, \gamma_u) = (400, 4600, 50, 50)$. The vertical gray dotted line indicates the initiation of pseudo-labeling in our method. Our CPG can generate pseudo-labels with a lower error rate and comparable utilization rate, achieving superior testing accuracy compared to both FreeMatch and SimPro.

distribution in later training stages implicitly (See Fig. 5 in Appendix A). Theoretically, we prove that this optimization cycle can significantly reduce the generalization error under some conditions. Empirically, Fig. 1(c) shows the superior pseudo-label prediction ability of our method.

To further improve our CPG, we introduce a class-aware adaptive augmentation (CAA) module and an auxiliary branch. The CAA module enhances minority class representations by dynamically adjusting augmentation intensity based on class compactness. Specifically, for classes with higher compactness (indicating lower intra-class diversity), CAA applies a smaller augmentation radius during representation synthesis to refine decision boundaries more effectively. Meanwhile, the auxiliary branch maximizes data utilization by enforcing prediction consistency between different augmentation views of all labeled and unlabeled samples.

In summary, the main contributions of this work are as follows.

- We propose a Controllable Pseudo-label Generation (CPG) framework to handle the unknown arbitrary unlabeled data distribution problem in ReaLTSSL. The framework expands the labeled dataset with the progressively identified reliable pseudo-labels from the unlabeled dataset, and trains the model on the updated labeled dataset with a known distribution, making it unaffected by the unlabeled data distribution.

- We theoretically prove that our controllable self-reinforcing optimization cycle can significantly reduce the generalization error under some conditions.

- We further propose a class-aware adaptive augmentation module to enhance the representation of minority classes, and an auxiliary branch to maximize data utilization by leveraging all labeled and unlabeled samples.

- Comprehensive experiments on four commonly used benchmarks (CIFAR-10-LT, CIFAR-100-LT, Food-101-LT, and SVHN-LT) under various labeled and unlabeled data distributions validate that our CPG achieves new state-of-the-art performances with a **15.97%** maximum improvement.

## 2 Related Work

**Long-Tailed Learning (LTL)** addresses the challenge of training effective models on data with long-tailed distributions, where a majority of classes are represented by only a few samples. Prevailing LTL methods can be categorized into three groups: re-sampling, logit adjustment, and ensemble learning. Re-sampling [17, 18, 19, 20] balances classes by under-sampling majority classes or over-sampling minority classes. Logit adjustment [21, 22, 23] modifies the predicted logits to compensate for class imbalance. Ensemble learning [24, 25, 26] combines multiple classifiers (experts) to improve the performance and robustness of the model. Despite their success in supervised learning, extending them to long-tailed semi-supervised learning (LTSSL) requires extra effort, as exploiting unlabeled samples is not trivial.

**Semi-Supervised Learning (SSL)** provides a powerful framework for improving model performance in label-scarce domains by leveraging abundant unlabeled data. Early SSL methods rely on self-

training with pseudo-labeling, where model refinement depends on the quality of generated pseudo-labels. Recent advances integrate pseudo-labeling [5, 6] with consistency regularization [27, 28], enforcing prediction invariance between weak and strong augmentation views to enhance robustness. Prominent examples include FixMatch [7], FlexMatch [29], FreeMatch [8], SoftMatch [9], and SemiReward [30], which primarily differ in their confidence thresholding strategies to balance pseudo-label quality and quantity. While effective in settings with balanced labeled and unlabeled data distributions, these methods struggle with long-tailed labeled data or mismatched distributions between labeled and unlabeled data.

**Long-Tailed Semi-Supervised Learning (LTSSL)** addresses the challenges arising from the long-tailed distribution. Existing LTSSL methods integrate long-tailed learning techniques [31, 32, 33, 34, 35] with frameworks like FixMatch. For example, ABC [10] employs an auxiliary classifier to improve minority class generalization, CReST [36] selectively generates pseudo-labels for minority class samples based on class frequency, and CoSSL [11] introduces feature enhancement strategies to refine classifier learning. However, these approaches overlook distribution mismatches between labeled and unlabeled data, a key limitation addressed by realistic long-tailed semi-supervised learning (ReaLTSSL). Recent ReaLTSSL methods include ACR [14], which adjusts logits based on the distance between the estimated unlabeled data distribution and the predefined anchor distributions; CPE [15], which employs multiple classifiers (experts) to model diverse unlabeled data distributions; Meta-Expert [37], which significantly enhances CPE by incorporating the expertises of different experts, and SimPro [13], which estimates unlabeled data distributions through high-confidence pseudo-label counts and adapts pseudo-label generation probabilities correspondingly. *Despite their contributions, ACR, CPE, and Meta-Expert require prior knowledge of the unlabeled data distribution. SimPro, meanwhile, fails to estimate the unlabeled data distribution accurately when the labeled and unlabeled distributions diverge significantly. These limitations restrict their practical applicability.*

## 3 Proposed Method

### 3.1 Preliminaries

**Problem formulation and challenges.** In ReaLTSSL, we consider a labeled dataset $\mathcal{D}_l = \{x_i^l, y_i^l\}_{i=1}^N$ of size $N$ and an unlabeled dataset $\mathcal{D}_u = \{x_j^u\}_{j=1}^M$ of size $M$, where both datasets share identical feature and label spaces. Here, $x_i^l$ denotes the $i$-th labeled sample with ground-truth label $y_i^l \in \{1, \ldots, C\}$, $x_j^u$ represents the $j$-th unlabeled sample, and $C$ is the total number of classes. Let $N_c$ denote the number of labeled samples in class $c$, defining the labeled data imbalance ratio as $\gamma_l = \frac{\max_c N_c}{\min_c N_c}$. While we can theoretically assume that $M_c$ represents the number of samples for class $c$ in $\mathcal{D}_u$, with its imbalance ratio $\gamma_u = \frac{\max_c M_c}{\min_c M_c}$, in practice the unlabeled data follow an unknown arbitrary distribution, making $M_c$ and $\gamma_u$ unavailable. The goal of ReaLTSSL is to train a classifier $F : \mathbb{R}^d \to [1, \ldots, C]$, parameterized by $\theta$, using $\mathcal{D}_l$ and $\mathcal{D}_u$ to achieve robust generalization across all classes. This presents a key challenge for SSL and LTSSL methods, as most existing methods depend on either balanced or aligned distributions between $\mathcal{D}_l$ and $\mathcal{D}_u$, or require prior knowledge of the unlabeled data distribution. In practice, the distribution of $\mathcal{D}_u$ is typically unknown, limiting the capability of the existing methods.

**Logit adjustment.** In real-world scenarios, data typically follow a long-tailed distribution, resulting in models trained on such data exhibiting an inherent bias toward majority classes. To mitigate this issue, numerous long-tailed learning methods aim to enhance minority class performance without degrading accuracy in majority classes. A prominent approach, logit adjustment [16], promotes larger relative margins between the minority positive and majority negative classes. It has been theoretically established as the Bayes-optimal solution for long-tailed learning. The logit-adjusted softmax cross-entropy loss (LA loss, $\ell_{la}$) is defined as:

$$\ell_{la}(y, f(g(x))) = -\log \frac{e^{f_y(g(x)) + \ln P_y}}{\sum_{y' \in [C]} e^{f_{y'}(g(x)) + \ln P_{y'}}}, \tag{1}$$

where $g(\cdot)$ and $f(\cdot)$ denote the encoder and classifier, respectively, $f_y(g(x))$ is the predicted logit of the $y$-th class, and $P_y$ denotes the class prior of the $y$-th class.

## 3.2 Our CPG Framework

**Framework overview.** To address the above-mentioned challenges, we introduce CPG, a controllable pseudo-label generation framework tailored for ReaLTSSL that operates *without* restrictive assumptions or prior knowledge about the distribution of the unlabeled dataset. Specifically, CPG adopts a controllable self-reinforcing optimization cycle, a class-aware adaptive augmentation module, and an auxiliary branch. The first employs a dynamic controllable filtering mechanism to identify reliable pseudo-labels. These pseudo-labels are combined with the original labeled dataset to iteratively update the labeled dataset, and progressively construct a Bayes-optimal classifier through logit adjustment. The improved classifier, in turn, helps identify more reliable pseudo-labels for subsequent training steps. The second enhances discriminative representations for minority classes within each training step. The last maximizes data utilization by leveraging all labeled and unlabeled samples. *Unlike existing methods that estimate the unlabeled data distribution based on high-confidence pseudo-labels, our CPG incorporates reliable pseudo-labels from the unlabeled dataset into the labeled dataset, and trains the model on the updated labeled dataset with a known distribution, making it unaffected by the unlabeled data distribution.*

**Controllable self-reinforcing optimization cycle.** The optimization cycle incorporates three key components, i.e., dynamic controllable filtering, iterative labeled dataset construction, and Bayes-optimal classifier construction. We present their details in the following paragraphs.

*Dynamic controllable filtering to identify reliable pseudo-labels.* Conventional SSL and LTSSL methods typically generate pseudo-labels by propagating predictions from weak augmentation views to their strong augmentation views. Although this strategy improves model robustness, it inherently introduces more error pseudo-labels when handling the unknown arbitrary unlabeled data distribution, as evidenced in Figs. 1(a) and 1(b). To overcome these limitations, we propose to identify reliable pseudo-labels to extend the labeled dataset instead of enforcing prediction consistency between weak and strong augmentation views. For each unlabeled sample $x_u$ with weak augmentation view $\Omega_w(x_u)$ and strong augmentation view $\Omega_s(x_u)$, we derive predicted pseudo-labels and corresponding confidence scores as:

$$\hat{q}_w, \tilde{q}_w = \arg\max(\sigma(f(h_w))), \max(\sigma(f(h_w))),$$
$$\hat{q}_s, \tilde{q}_s = \arg\max(\sigma(f(h_s))), \max(\sigma(f(h_s))), \tag{2}$$

where $\arg\max(\cdot)$ and $\max(\cdot)$ extract the pseudo-labels and confidence scores, respectively, and $\sigma(\cdot)$ denotes the softmax function. Let $h_w = g(\Omega_w(x_u))$ represent the representation produced by $g(\cdot)$ from $\Omega_w(x_u)$, with $\hat{q}_w$ and $\tilde{q}_w$ denoting the pseudo-label and its associated confidence score predicted for $h_w$, respectively. $h_s$, $\hat{q}_s$, and $\tilde{q}_s$ are defined similarly for $\Omega_s(x_u)$. We express the binary sample mask for selecting a reliable pseudo-label as:

$$\mathbb{I} = \mathbb{I}(\tilde{q}_w > \tau) \cdot \mathbb{I}(\tilde{q}_s > \tau) \cdot \mathbb{I}(\hat{q}_w = \hat{q}_s), \tag{3}$$

where $\tau$ denotes the confidence threshold and $\cdot$ represents element-wise multiplication. Since samples may receive conflicting pseudo-labels across different training steps according to Eq. (3), we introduce a voting strategy to ensure consistent pseudo-label assignments for each sample. As illustrated in Fig. 1(c), the combination of Eq. (3) and the voting strategy effectively controls pseudo-label generation and enhances pseudo-label reliability, demonstrating its strong performance.

*Iterative labeled dataset construction.* During the training process, we maintain both a labeled dataset and a pseudo-labeled dataset (i.e., identified reliable pseudo-labels) at each training step, with class frequencies denoted by $n = \{n_1, \dots, n_C\}$ and $m = \{m_1, \dots, m_C\}$, respectively. The updated labeled dataset's class frequency is defined as $\phi = \{\phi_1, \dots, \phi_C\}$, where $\phi_c = n_c + m_c$ denotes the number of samples for class $c$ in the updated labeled dataset. Accordingly, the class distribution for the updated labeled dataset is given by:

$$\pi = \{\pi_1, \dots, \pi_C\}, \pi_c = \frac{\phi_c}{\sum_{c' \in [C]} \phi_{c'}}, \tag{4}$$

where $\pi_c$ represents the label frequency of the $c$-th class in the updated labeled dataset. This yields an updated labeled dataset with a known distribution at each training step, enabling the iterative construction of a Bayes-optimal classifier.

*Bayes-optimal classifier construction.* Leveraging the known distribution of the updated labeled dataset, we construct a classifier by minimizing the LA loss ($\ell_{la}$) defined in Eq. (1). The resulting classifier minimizes the balanced error and yields Bayes-optimal predictions [16]. By iteratively

constructing a Bayes-optimal classifier using the updated labeled dataset (with a known distribution at each training step), our CPG framework generates increasingly reliable pseudo-labels in subsequent training steps. Although our method does not explicitly estimate the unlabeled data distribution, the progressive incorporation of reliable pseudo-labels naturally approximates the ground-truth unlabeled data distribution as training progresses. This is because the model's increasing confidence enables it to gradually include more diverse samples, including those from minority classes, improving distribution alignment (as demonstrated in Appendix A Fig. 5).

While consistency regularization between augmentation views is commonly used in SSL, we avoid it in our controllable self-reinforcing optimization cycle for two key reasons. First, pseudo-label quality is highly sensitive to the distribution mismatch between labeled and unlabeled data in LTSSL. Second, propagating error pseudo-labels from weak augmentation views to strong augmentation views can reinforce the model's learning of incorrect predictions, leading to error accumulation and performance degradation (as demonstrated in Figs. 1 and 3).

**Class-aware adaptive augmentation to enhance minority class representations.** While the controllable self-reinforcing optimization cycle effectively utilizes unlabeled samples with reliable pseudo-labels and iteratively constructs Bayes-optimal classifiers, enhancing discriminative representations for minority classes remains crucial in long-tailed learning. To achieve this goal, we enhance the updated labeled dataset by synthesizing minority class representations guided by class compactness. Formally, the compactness of class $c$ is defined as:

$$\alpha(c) = \frac{1}{\phi_c} \sum_{i \in [\phi_c]} \frac{\langle h_i, \mu(c) \rangle}{\|h_i\| \cdot \|\mu(c)\|}, \tag{5}$$

where $\phi_c$ denotes the number of samples in class $c$, $h_i = g(x_i)$ denotes the representation produced by encoder $g(\cdot)$, $\mu(c)$ denotes the centroid of class $c$ in the feature space, and $\|\cdot\|$ denotes the Euclidean norm. As minority classes typically exhibit lower intra-class diversity, it is associated with a higher compactness $\alpha$ according to Eq. (5) and benefits from representation synthesis with a smaller augmentation radius $r$ to refine decision boundaries, where $r = 1/\alpha$. The representation synthesis for sample $x_i$ from minority class $c$ is computed as:

$$h_i' = h_i + \frac{h_i}{\|h_i\|} \cdot r(c) \cdot \delta_i, \tag{6}$$

where $\delta_i$ is a noise vector sampled from a $d$-dimensional standard normal distribution (with $d$ matching the representation dimension). For minority classes in the updated labeled dataset, we synthesize ten augmented representations per original sample by Eq. (6).

**Auxiliary branch to maximize data utilization.** Our controllable self-reinforcing optimization cycle incorporates reliable pseudo-labels from the unlabeled dataset into the labeled dataset and trains the model on the updated labeled dataset. However, during early training stages when model performance is still weak, only a limited number of samples meet the reliability condition for pseudo-labeling (as demonstrated in Figs. 1 and 3, and Appendix A Fig. 5). To maximize data utilization and provide additional supervision signals to stabilize representation learning during the critical initial phase, we introduce an auxiliary branch to leverage all labeled and unlabeled samples. Specifically, this branch adopts a consistency regularization paradigm similar to the conventional SSL methods like FixMatch [7], enforcing prediction consistency between weak and strong augmentation views via an unsupervised consistency regularization loss (Aux loss, $\ell_{aux}$). The Aux loss $\ell_{aux}$ is formulated as:

$$\ell_{aux} = -\log \frac{e^{f_{\hat{y}}(g(x))}}{\sum_{y' \in [C]} e^{f_{y'}(g(x))}}, \tag{7}$$

where $f_{\hat{y}}(g(x))$ denotes the logit produced by the strong augmentation view, corresponding to the pseudo-label $\hat{y}$ generated from the weak augmentation view.

### 3.3 Model Training and Prediction

Our training process consists of an initial stage of thirty epochs using only the labeled dataset, followed by an iterative stage that expands the labeled dataset with reliable pseudo-labels and continues optimization on the updated labeled dataset. Throughout both stages, the auxiliary branch is trained on all labeled and unlabeled samples. We optimize the model by combining the LA loss ($\ell_{la}$) applied to both branches, with the Aux loss ($\ell_{aux}$) applied only to the auxiliary branch. The overall loss $\ell_{overall}$ is formulated as:

---

**Algorithm 1** Training process of our CPG

---

1: **Input**: Labeled and unlabeled datasets $\mathcal{D}_l$ and $\mathcal{D}_u$.
2: **Output**: Encoder $g$, primary branch (classifier) $f_{pri}$, and auxiliary branch (classifier) $f_{aux}$.
3: Initialize the parameters of $g$, $f_{pri}$, and $f_{aux}$ randomly.
4: **for** epoch=1, 2, . . . **do**
5:    **for** batch=1, 2, . . . **do**
6:       **if** epoch $> 30$ **then**
7:          Identify reliable pseudo-labels by Eq. (2) and Eq. (3);
8:          Calculate the class distribution of the updated labeled dataset by Eq. (4);
9:          Enhance minority class representations by Eq. (5) and Eq. (6);
10:       **end if**
11:       Calculate the LA loss by Eq. (1) for the primary branch;
12:       Calculate the LA loss by Eq. (1) and Aux loss by Eq. (7) for the auxiliary branch;
13:       Obtain the overall loss by Eq. (8);
14:       Update network parameters via gradient descent;
15:    **end for**
16: **end for**

---

$$\ell_{overall} = \ell_{la} + \omega \ell_{aux}, \tag{8}$$

where $\omega$ is a binary indicator function that equals 1 when $\ell_{overall}$ is applied to the auxiliary branch and 0 otherwise. The pseudo-code is detailed in Algorithm 1. After training, we can obtain the predicted label of an unseen sample $x^*$ by getting the label index with the highest confidence.

### 3.4 Theoretical Analysis

We prove that our controllable self-reinforcing optimization cycle (including progressively incorporating reliable pseudo-labels into the labeled dataset and training a Bayes-optimal classifier on the updated labeled dataset with a known distribution) can significantly reduce the generalization error under some conditions. Let $\ell_{ours} = \frac{1}{N+\hat{M}_t} \left[ \sum_{i=1}^{N} \ell_{la}(f(x_i), y_i) + \sum_{j=1}^{\hat{M}_t} \ell_{la}(f(x_j), \hat{y}_j) \right]$ be the training loss on the $N$ labeled and $\hat{M}_t$ pseudo-labeled samples with pseudo-labeling error rate $\epsilon_t$ for our method at the $t$-th training step. We begin by defining the model empirical risk $R$ at the $t$-th training step as:

$$R_t = R_{t-1} - \lambda_t + I_{\epsilon_t} + I_{O_t}, \tag{9}$$

where $R_t$ denotes the model empirical risk at the $t$-th training step, $\lambda_t \geq 0$ quantifies the empirical risk reduction achieved, and $I_{\epsilon_t}$ and $I_{O_t}$ represent the impact of noisy pseudo-labels and number of training samples, respectively. Let the function space $\mathcal{H}_y$ for the label $y \in \{1, \ldots, C\}$ be $\{h : x \longmapsto f_y(x) | f \in \mathcal{F}\}$, where $f_y(x)$ denotes the predicted probability of the $y$-th class. Let $\mathcal{R}_{O_t}(\mathcal{H}_y)$ be the expected Rademacher complexity [38] of $\mathcal{H}_y$ with $O_t = N + \hat{M}_t$ training samples. Then we have the following theorem.

**Theorem 1** (Generalization Error). *Suppose that the loss function $\ell_{la}(f(x), y)$ is $\rho$-Lipschitz with respect to $f(x)$ for all $y \in \{1, \ldots, C\}$ and upper-bounded by $U$. Given the pseudo-labeling error rate $0 < \epsilon < 1$, for any $\upsilon > 0$, with probability at least $1 - \upsilon$, we have:*

$$R_T \leq R_0 - \sum_{t=1}^{T} \lambda_t + U \sum_{t=1}^{T} \epsilon_t + 4\sqrt{2}\rho \sum_{t=1}^{T} \sum_{y=1}^{C} \mathcal{R}_{O_t}(\mathcal{H}_y) + 2U \sum_{t=1}^{T} \sqrt{\frac{\log \frac{2}{\upsilon}}{2O_t}}, \tag{10}$$

*where $T$ denotes the total number of training steps, $R_T$ and $R_0$ represent the model empirical risk at the final training step and initial training step (without pseudo-labels), respectively, and $\epsilon_t$ represents the pseudo-labeling error rate at the $t$-th training step.*

The proof of Theorem 1 is provided in Appendix B. It can be observed that the model empirical risk $R_T$ depends primarily on three key factors, i.e., the cumulative model empirical risk reduction $\sum_{t=1}^{T} \lambda_t$, the cumulative pseudo-labeling error rate $\sum_{t=1}^{T} \epsilon_t$, and the number of training samples $O_t$ at each training step. When the training step $t$ increases, the number of training samples $O_t$ grows while the pseudo-labeling error rate $\epsilon_t$ either decreases or remains constant, *the Bayes-optimal*

*classifier constructed at the $t$-th training step can maximize the model empirical risk reduction $\lambda_t$, i.e.,* as $T \to \infty$, $O_T \to \infty$, $\sum_{t=1}^{T} \epsilon_t$ remains bounded, and $\sum_{t=1}^{T} \lambda_t$ reaches its theoretical maximum, Theorem 1 demonstrates that the generalization error is minimized. As shown in Figs. 3(a) and 3(b), our CPG framework demonstrates superior performance over baseline methods, achieving both: (i) a consistently lower pseudo-labeling error rate $\epsilon_t$, and (ii) a growing number of training samples $O_t$ that gradually match baseline levels during training. Together with the Bayes-optimal classifier, our CPG can significantly reduce the generalization error.

## 4 Experiments

### 4.1 Experimental setting

**Dataset.** We perform our experiments on four widely-used datasets (CIFAR-10-LT [39], CIFAR-100-LT [39], Food-101-LT [40], and SVHN-LT [41]), following the main experimental settings in FreeMatch [8] and SimPro [13]. More details about those datasets are provided in Appendix C, and implementation details are provided in Appendix D.

**Baselines.** We compare with three SSL algorithms, including FixMatch [7], FreeMatch [8], and SoftMatch [9], and three ReaLTSSL algorithms, including ACR [14], SimPro [13], and CDMAD [42]. Moreover, we use the supervised learning (SL) setting as a performance upper-bound reference. For a fair comparison, we test these baselines and our CPG on the widely-used codebase USB[2]. For the data augmentation strategy, an identical weak augmentation is applied to both labeled and unlabeled data, while reserving a strong augmentation for unlabeled data. We use the same dataset splits with no overlap between labeled and unlabeled training data for all datasets.

Table 1: Comparison of accuracy (%) on CIFAR-10-LT ($N_{max} = 400$, $M_{max} = 4600$) with class imbalance ratio $\gamma \in \{100, 150, 200\}$ under different unlabeled data distributions. CE and LA denote using softmax cross-entropy loss and logit-adjusted softmax cross-entropy loss under the supervised learning (SL) setting, respectively. † indicates we reproduce ACR without anchor distributions for a fair comparison.

| Scenario | Method | $\gamma = 100$ | | | $\gamma = 150$ | | | $\gamma = 200$ | | | Avg. |
|---|---|---|---|---|---|---|---|---|---|---|---|
| | | Arbitrary | Inverse | Consistent | Arbitrary | Inverse | Consistent | Arbitrary | Inverse | Consistent | |
| SL | CE | 82.90 ±2.00 | 83.89 ±0.36 | 79.14 ±0.20 | 81.06 ±2.42 | 82.48 ±0.30 | 74.51 ±0.46 | 79.20 ±1.21 | 81.43 ±0.53 | 72.12 ±0.18 | 79.64 ±0.85 |
| | LA [16] | 87.00 ±0.65 | 86.84 ±0.36 | 85.92 ±0.14 | 85.61 ±0.96 | 85.59 ±0.09 | 83.88 ±0.55 | 84.55 ±0.71 | 85.02 ±0.38 | 82.56 ±0.22 | 85.22 ±0.45 |
| SSL | FixMatch [7] | 62.17 ±4.64 | 57.61 ±1.71 | 68.35 ±1.44 | 57.73 ±3.74 | 56.51 ±2.22 | 64.54 ±1.34 | 54.16 ±4.03 | 51.43 ±5.24 | 61.53 ±1.08 | 59.34 ±2.83 |
| | FreetMatch [8] | 65.41 ±3.93 | 68.91 ±1.78 | 70.08 ±0.67 | 63.35 ±4.84 | 66.69 ±2.77 | 63.47 ±2.30 | 62.35 ±6.30 | 64.31 ±4.44 | 59.53 ±1.50 | 64.90 ±3.17 |
| | SoftMatch [9] | 66.13 ±2.27 | 66.00 ±1.56 | 72.75 ±1.09 | 62.92 ±2.22 | 65.57 ±0.53 | 68.85 ±1.28 | 59.09 ±1.95 | 63.62 ±2.13 | 62.49 ±0.48 | 65.27 ±1.50 |
| ReaLTSSL | ACR† [14] | 60.60 ±4.21 | 62.54 ±4.04 | 73.20 ±1.80 | 48.22 ±8.07 | 51.08 ±1.03 | 68.31 ±0.35 | 50.13 ±7.50 | 53.40 ±3.49 | 65.12 ±1.05 | 59.18 ±3.51 |
| | SimPro [13] | 65.81 ±1.60 | 63.70 ±1.50 | 64.13 ±2.03 | 61.41 ±8.10 | 61.27 ±0.72 | 62.32 ±2.41 | 59.11 ±8.49 | 58.09 ±3.74 | 59.31 ±3.11 | 61.68 ±3.52 |
| | CDMAD [42] | 63.39 ±1.34 | 60.24 ±3.13 | 64.63 ±1.57 | 59.45 ±1.17 | 61.44 ±1.11 | 65.29 ±0.93 | 56.39 ±2.45 | 58.68 ±3.90 | 59.38 ±1.85 | 60.99 ±1.94 |
| | Ours | **82.10** ±0.74 | **82.37** ±0.15 | 76.93 ±2.68 | **76.38** ±3.87 | **78.19** ±1.63 | 70.75 ±2.13 | **75.18** ±3.28 | **77.45** ±1.44 | **68.33** ±3.94 | **76.41** ±2.21 |

Table 2: Comparison of accuracy (%) on CIFAR-100-LT ($N_{max} = 50$, $M_{max} = 450$) with class imbalance ratio $\gamma \in \{10, 15, 20\}$ under different unlabeled data distributions.

| Scenario | Method | $\gamma = 10$ | | | $\gamma = 15$ | | | $\gamma = 20$ | | | Avg. |
|---|---|---|---|---|---|---|---|---|---|---|---|
| | | Arbitrary | Inverse | Consistent | Arbitrary | Inverse | Consistent | Arbitrary | Inverse | Consistent | |
| SL | CE | 64.73 ±0.27 | 65.35 ±0.21 | 64.62 ±0.24 | 61.85 ±0.29 | 63.12 ±0.26 | 61.61 ±0.26 | 59.94 ±0.33 | 61.29 ±0.27 | 58.91 ±0.29 | 62.38 ±0.27 |
| | LA [16] | 66.83 ±0.33 | 66.68 ±0.18 | 66.50 ±0.28 | 64.52 ±0.47 | 64.89 ±0.25 | 63.80 ±0.21 | 62.73 ±0.38 | 63.44 ±0.30 | 62.22 ±0.35 | 64.62 ±0.30 |
| SSL | FixMatch [7] | 48.89 ±1.27 | 47.92 ±1.63 | 50.04 ±0.80 | 42.91 ±0.64 | 41.54 ±1.00 | 44.60 ±0.48 | 40.64 ±1.06 | 38.77 ±0.60 | 42.11 ±0.57 | 44.16 ±0.89 |
| | FreetMatch [8] | 45.97 ±0.57 | 45.74 ±1.35 | 46.36 ±0.93 | 40.66 ±0.62 | 40.68 ±0.48 | 41.08 ±0.26 | 38.05 ±0.12 | 39.24 ±0.27 | 39.61 ±0.80 | 41.93 ±0.60 |
| | SoftMatch [9] | 47.99 ±0.85 | 48.17 ±1.04 | 48.86 ±0.34 | 42.62 ±1.34 | 41.42 ±0.46 | 43.78 ±0.84 | 40.24 ±0.93 | 40.17 ±1.10 | 41.13 ±0.35 | 43.82 ±0.80 |
| ReaLTSSL | ACR† [14] | 44.10 ±1.59 | 48.50 ±0.86 | 41.33 ±0.41 | 35.12 ±1.36 | 39.01 ±0.38 | 32.73 ±1.74 | 29.08 ±1.11 | 32.78 ±1.59 | 29.69 ±2.94 | 36.93 ±1.33 |
| | SimPro [13] | 44.26 ±0.57 | 44.14 ±1.12 | 45.67 ±0.88 | 39.00 ±0.50 | 37.00 ±0.39 | 41.32 ±0.80 | 35.57 ±1.11 | 34.35 ±0.63 | 37.86 ±0.65 | 39.91 ±0.74 |
| | CDMAD [42] | 33.95 ±2.94 | 35.02 ±1.62 | 37.15 ±0.95 | 31.36 ±1.55 | 29.60 ±4.17 | 32.62 ±4.81 | 28.89 ±5.23 | 25.18 ±1.68 | 28.37 ±1.58 | 31.35 ±2.72 |
| | Ours | **51.48** ±0.32 | **51.46** ±0.22 | **51.22** ±0.71 | **46.26** ±1.40 | **47.88** ±0.24 | **45.94** ±1.02 | **44.17** ±1.30 | **44.17** ±1.01 | **42.66** ±0.56 | **47.25** ±0.75 |

### 4.2 Results

**Consistent distribution settings.** We begin our analysis by evaluating performance under the consistent distribution setting, where labeled and unlabeled data follow an identical long-tailed distribution. Tables 1 and 2 present the primary results for CIFAR-10-LT and CIFAR-100-LT, respectively. Across all imbalance ratios, CPG achieves superior classification accuracy compared to previous baselines. For example, given $(N_{max}, M_{max}, \gamma) = (400, 4600, 100)$, CPG outperforms all baselines by up to 3.73 percentage points (pp) on CIFAR-10-LT. Similar improvements are observed for Food-101-LT and SVHN-LT in Table 3. On Food-101-LT with $(N_{max}, M_{max}, \gamma) = (50, 450, 15)$, CPG achieves gains of up to 2.27 pp.

---

[2]https://github.com/microsoft/Semi-supervised-learning

Table 3: Comparison of accuracy (%) on Food-101-LT ($N_{max} = 50, M_{max} = 450$) with class imbalance ratio $\gamma \in \{10, 15\}$ and SVHN-LT ($N_{max} = 400, M_{max} = 4600, \gamma = 100$) under different unlabeled data distributions.

| Scenario | Method | Food-101-LT | | | | | | Avg. | SVHN-LT | | | Avg. |
|---|---|---|---|---|---|---|---|---|---|---|---|---|
| | | $\gamma = 10$ | | | $\gamma = 15$ | | | | $\gamma = 100$ | | | |
| | | Arbitrary | Inverse | Consistent | Arbitrary | Inverse | Consistent | | Arbitrary | Inverse | Consistent | |
| SL | CE | 48.51 ±0.73 | 49.44 ±0.41 | 47.92 ±0.35 | 45.45 ±1.16 | 46.40 ±0.08 | 44.58 ±0.29 | 47.05 ±0.50 | 91.58 ±0.60 | 92.19 ±0.29 | 91.92 ±0.10 | 91.90 ±0.33 |
| | LA [16] | 50.56 ±0.26 | 50.60 ±0.30 | 50.06 ±0.35 | 47.78 ±0.75 | 48.32 ±0.16 | 47.36 ±0.25 | 49.11 ±0.35 | 93.26 ±0.90 | 94.37 ±0.09 | 92.13 ±0.40 | 93.25 ±0.46 |
| SSL | FixMatch [7] | 22.28 ±0.88 | 21.18 ±0.30 | 23.37 ±0.49 | 18.46 ±1.07 | 17.47 ±0.40 | 19.77 ±0.24 | 20.42 ±0.56 | 89.57 ±2.34 | 87.29 ±0.37 | 93.13 ±0.04 | 90.00 ±0.92 |
| | FreetMatch [8] | 22.28 ±1.12 | 21.89 ±0.87 | 22.22 ±0.68 | 18.72 ±0.80 | 18.66 ±0.46 | 19.61 ±0.38 | 20.56 ±0.72 | 85.78 ±2.87 | 87.01 ±1.56 | 90.88 ±0.48 | 87.89 ±1.64 |
| | SoftMatch [9] | 22.87 ±0.76 | 22.30 ±0.73 | 22.72 ±0.75 | 19.46 ±1.04 | 19.43 ±0.87 | 19.84 ±0.94 | 21.10 ±0.85 | 85.71 ±1.87 | 87.79 ±1.15 | 91.07 ±0.42 | 88.19 ±1.15 |
| ReaLTSSL | ACR† [14] | 18.41 ±0.75 | 19.27 ±1.12 | 17.30 ±0.58 | 15.44 ±0.50 | 17.07 ±0.61 | 13.88 ±0.73 | 16.90 ±0.72 | 87.67 ±4.60 | 84.20 ±1.58 | 92.64 ±0.50 | 88.17 ±2.23 |
| | SimPro [13] | 19.59 ±0.77 | 17.31 ±0.50 | 21.34 ±0.41 | 15.29 ±0.93 | 14.51 ±0.72 | 17.63 ±0.42 | 17.61 ±0.63 | 90.10 ±2.75 | 88.50 ±0.12 | 86.55 ±4.32 | 88.38 ±2.40 |
| | CDMAD [42] | 10.11 ±1.63 | 11.26 ±1.57 | 12.24 ±1.09 | 10.21 ±1.43 | 8.65 ±1.60 | 11.18 ±1.64 | 10.61 ±1.49 | 81.90 ±1.81 | 81.39 ±3.46 | 87.88 ±3.92 | 83.72 ±3.06 |
| | Ours | **25.98** ±0.66 | **25.52** ±0.43 | **25.24** ±0.30 | **21.88** ±0.85 | **21.27** ±0.51 | **22.11** ±0.72 | **23.67** ±0.58 | **93.99** ±0.34 | **94.74** ±0.49 | **93.45** ±0.52 | **94.06** ±0.45 |

**Inconsistent distribution settings.** To assess the real-world scenarios with mismatched distributions, we evaluate CPG under inverse long-tailed and arbitrary unlabeled data distributions. As shown in Tables 1 and 2, CPG consistently surpasses baselines on both CIFAR-10-LT and CIFAR-100-LT. Notably, for $(N_{max}, M_{max}, \gamma) = (400, 4600, 100)$ with arbitrary unlabeled data distribution, CPG achieves performance gains of **15.97 pp** over existing methods on CIFAR-10-LT. For realistic benchmark Food-101-LT in Table 3, CPG outperforms all competitors by up to 3.22 pp given the setting $(N_{max}, M_{max}, \gamma) = (50, 450, 10)$ with inverse long-tailed unlabeled data distribution.

When evaluating the overall performance, our CPG framework consistently outperforms baselines across all datasets. Notably, it achieves an average accuracy improvement of **11.14 pp** on CIFAR-10-LT, 3.09 pp on CIFAR-100-LT, and 4.06 pp on SVHN-LT. Moreover, it maintains a significant gain of 2.57 pp on the challenging realistic benchmark Food-101-LT.

Note that we observe a less significant performance improvement on CIFAR-100-LT than on CIFAR-10-LT. This can be attributed to the inherent challenges of CIFAR-100. Although CIFAR-100 and CIFAR-10 have the same total training set size, CIFAR-100 divides the data into 100 fine-grained classes (compared to 10 in CIFAR-10), reducing per-class samples by an order of magnitude. The finer class granularity also increases inter-class similarity, making classification inherently more challenging. As a result, the upper-bound performance (e.g., supervised learning) on CIFAR-100-LT (64.62 pp) is significantly lower than on CIFAR-10-LT (85.22 pp). Moreover, our method leverages consistency regularization to enhance model stability and prevent severe performance degradation, maintaining a relatively stable lower-bound performance. The narrower gap between this lower bound and the constrained upper bound on CIFAR-100-LT inherently limits the absolute performance gains. Despite this, our method still achieves a substantial average accuracy improvement of 3.09 pp on CIFAR-100-LT, as detailed in Table 2.

### 4.3 Analysis

**Evaluation under varying imbalance ratios.** Fig. 4 illustrates the performance of CPG across different unlabeled data imbalance ratios, where the labeled data distribution is long-tailed with a fixed imbalance ratio and the unlabeled data distribution is arbitrary with varying imbalance ratios. The datasets are CIFAR-10-LT with ($\gamma_l = 100$, $\gamma_u \in \{100, 150, 200, 250\}$) and CIFAR-100-LT with ($\gamma_l = 10$, $\gamma_u \in \{5, 10, 15, 20\}$). The results consistently indicate that our method outperforms previous baselines. Moreover, as the unlabeled data imbalance ratio increases, the performance of baseline methods degrades significantly. In contrast, our method maintains nearly consistent performance on CIFAR-10-LT and exhibits only minor degradation on CIFAR-100-LT, demonstrating robust performance across varying unlabeled data imbalance ratios.

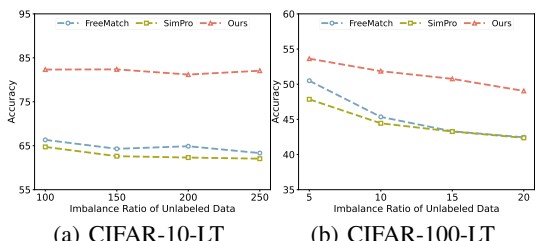

(a) CIFAR-10-LT  (b) CIFAR-100-LT

Figure 4: Comparison of accuracy (%) among FreeMatch [8], SimPro [13], and our CPG on CIFAR-10-LT and CIFAR-100-LT under arbitrary unlabeled data distribution with long-tailed labeled data distribution.

**Evaluation under arbitrary labeled data distribution.** Real-world scenarios often exhibit dynamic distribution shifts where any class can become dominant, affecting both labeled and unlabeled data. While existing evaluations assume static long-tailed distributions for labeled data, we assess

CPG under arbitrary labeled data distributions, with unlabeled data following either a consistent or inconsistent distribution relative to labeled data. As shown in Table 4, CPG consistently outperforms baselines on CIFAR-10-LT and CIFAR-100-LT, demonstrating exceptional robustness. For instance, with $(N_{max}, M_{max}, \gamma) = (400, 4600, 100)$ and an inconsistent unlabeled data distribution, CPG achieves performance gains of **11.82 pp** over existing methods on CIFAR-10-LT.

Table 4: Comparison of accuracy (%) on CIFAR-10-LT and CIFAR-100-LT under arbitrary labeled and unlabeled data distributions.

| Scenario | Method | CIFAR-10-LT | | CIFAR-100-LT | |
|---|---|---|---|---|---|
| | | $(N_{max}, M_{max}, \gamma) = (400, 4600, 100)$ | | $(N_{max}, M_{max}, \gamma) = (50, 450, 10)$ | |
| | | Inconsistent | Consistent | Inconsistent | Consistent |
| SL | CE | 82.90 ±3.34 | 78.48 ±4.61 | 64.75 ±0.58 | 63.77 ±0.31 |
| | LA [16] | 87.22 ±1.29 | 85.11 ±1.63 | 66.54 ±0.29 | 66.35 ±0.37 |
| SSL | FixtMatch [7] | 66.27 ±3.69 | 69.69 ±3.50 | 48.23 ±0.80 | 48.59 ±0.92 |
| | FreetMatch [8] | 62.02 ±2.21 | 69.73 ±2.80 | 45.59 ±0.31 | 46.49 ±0.78 |
| | SoftMatch [9] | 63.37 ±1.68 | 73.07 ±2.79 | 47.79 ±0.24 | 48.83 ±0.84 |
| ReaLTSSL | ACR† [14] | 58.15 ±5.03 | 71.71 ±0.74 | 45.30 ±0.56 | 42.34 ±0.28 |
| | SimPro [13] | 70.40 ±5.18 | 67.54 ±2.31 | 44.10 ±0.45 | 45.82 ±1.53 |
| | CDMAD [42] | 60.26 ±0.26 | 66.45 ±2.89 | 32.86 ±5.85 | 35.79 ±3.22 |
| | Ours | **82.22** ±4.04 | **76.41** ±4.13 | **51.59** ±1.06 | **50.59** ±0.83 |

**Ablation study.** We conduct ablation studies to validate the effectiveness of key components in our CPG in Table 5. Specifically, we set $(N_{max}, M_{max}, \gamma) = (400, 4600, 100)$ for CIFAR-10-LT and $(N_{max}, M_{max}, \gamma) = (50, 450, 10)$ for CIFAR-100-LT, respectively. As shown in Table 5, we can observe that **each component brings a significant improvement**. For example, on CIFAR-10-LT, the auxiliary branch (AB) brings a performance gain of 0.80 pp given the arbitrary unlabeled data distribution. Moreover, class-aware adaptive augmentation (CAA) module further boosts performance by 2.99 pp over the AB baseline, while controllable self-reinforcing optimization cycle (CSOC) delivers a more substantial gain of **14.41 pp**. Combining CAA and CSOC achieves a synergistic improvement of **14.82 pp**, underscoring the complementary roles of these components. When evaluating the overall performance, CAA and CSOC individually contribute average performance gains of 2.35 pp and 6.97 pp, respectively, while their combination yields a **10.65 pp** improvement, reinforcing the effectiveness and robustness of our design.

Table 5: Comparison of accuracy (%) on with and without the key components in the proposed method.

| Ablations | | | CIFAR-10-LT | | | CIFAR-100-LT | | | Avg. |
|---|---|---|---|---|---|---|---|---|---|
| w/ AB | w/ CSOC | w/ CAA | Arbitrary | Inverse | Consistent | Arbitrary | Inverse | Consistent | |
| | | | 65.18 | 63.94 | 66.10 | 46.32 | 46.23 | 45.84 | 55.60 |
| ✓ | | | 65.98 | 64.93 | 69.28 | 48.37 | 47.85 | 48.13 | 57.42 ↑ 1.82 |
| ✓ | | ✓ | 68.97 | 67.59 | 75.52 | 48.90 | 48.26 | 49.39 | 59.77 ↑ 2.35 |
| ✓ | ✓ | | 80.39 | 80.85 | 76.85 | 49.25 | 49.66 | 49.35 | 64.39 ↑ 6.97 |
| ✓ | ✓ | ✓ | **82.33** | **82.32** | **78.35** | **51.85** | **51.61** | **51.04** | **66.25** ↑10.65 |

**Statistical significance.** Table 6 presents the win/tie/loss counts between our CPG and six baseline methods across different datasets under varying unlabeled data distributions, using a pairwise t-test at a 0.05 significance level. The results show that our CPG outperforms baseline methods in all cases, with a significant improvement in 80.25 pp of cases (130 out of 162), demonstrating its effectiveness.

More experiment results and ablation studies are provided in Appendix E and Appendix F, respectively.

Table 6: Statistical significance of performance differences assessed with pairwise t-test at a 0.05 significance level, reported as win/tie/loss counts.

| Scenario | Method | Arbitrary | Inverse | Consistent | Total |
|---|---|---|---|---|---|
| SSL | FixtMatch [7] | 7 / 2 / 0 | 9 / 0 / 0 | 4 / 5 / 0 | 20 / 7 / 0 |
| | FreetMatch [8] | 9 / 0 / 0 | 9 / 0 / 0 | 7 / 2 / 0 | 25 / 2 / 0 |
| | SoftMatch [9] | 7 / 2 / 0 | 8 / 1 / 0 | 5 / 4 / 0 | 20 / 7 / 0 |
| ReaLTSSL | ACR† [14] | 8 / 1 / 0 | 8 / 1 / 0 | 5 / 4 / 0 | 21 / 6 / 0 |
| | SimPro [13] | 6 / 3 / 0 | 9 / 0 / 0 | 7 / 2 / 0 | 22 / 5 / 0 |
| | CDMAD [42] | 8 / 1 / 0 | 9 / 0 / 0 | 5 / 4 / 0 | 22 / 5 / 0 |
| | Total | 45 / 9 / 0 | 52 / 2 / 0 | 33 / 21 / 0 | 130 / 32 / 0 |

## 5 Conclusion

In this work, we have presented a novel solution to the ReaLTSSL problem via controllable pseudo-label generation. We proposed to expand the labeled dataset with reliable pseudo-labels and train the model on the updated labeled dataset with a known distribution, making it unaffected by the unlabeled data distribution. Our framework introduces a controllable self-reinforcing optimization cycle, combining dynamic controllable filtering for reliable pseudo-label identifying with iterative Bayes-optimal classifier construction through logit adjustment. It is further enhanced by a class-aware adaptive augmentation module for minority class representation learning and an auxiliary branch to maximize data utilization by leveraging all labeled and unlabeled samples. We further theoretically proved that this optimization cycle can significantly reduce the generalization error under some conditions. Comprehensive experiments on four commonly used benchmarks show consistent improvements of our method over the state-of-the-art methods by up to **15.97%** across various scenarios.

## Acknowledgments and Disclosure of Funding

This work was supported by the National Natural Science Foundation of China under Grants U24A20322, 62576094 and 62422118. This work is also supported by Hong Kong UGC under grants UGC/FDS11/E03/24, UGC/FDS11/E03/25, and Hong Kong Research Grants Council under Grant 11219324. This research work is also supported by the Big Data Computing Center of Southeast University.

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

# A Evolution of Pseudo-label Predictions

Fig. 5 shows the evolution of pseudo-label predictions of our method under the arbitrary unlabeled data distribution, evaluated on CIFAR-10-LT. From Fig. 5, we observe that our CPG progressively increases the utilization rate of pseudo-labels while maintaining the high accuracy. The predicted distribution gradually approximates the ground-truth unlabeled data distribution, and the Kullback-Leibler (KL) divergence between the predicted and ground-truth unlabeled data distributions decreases during training.

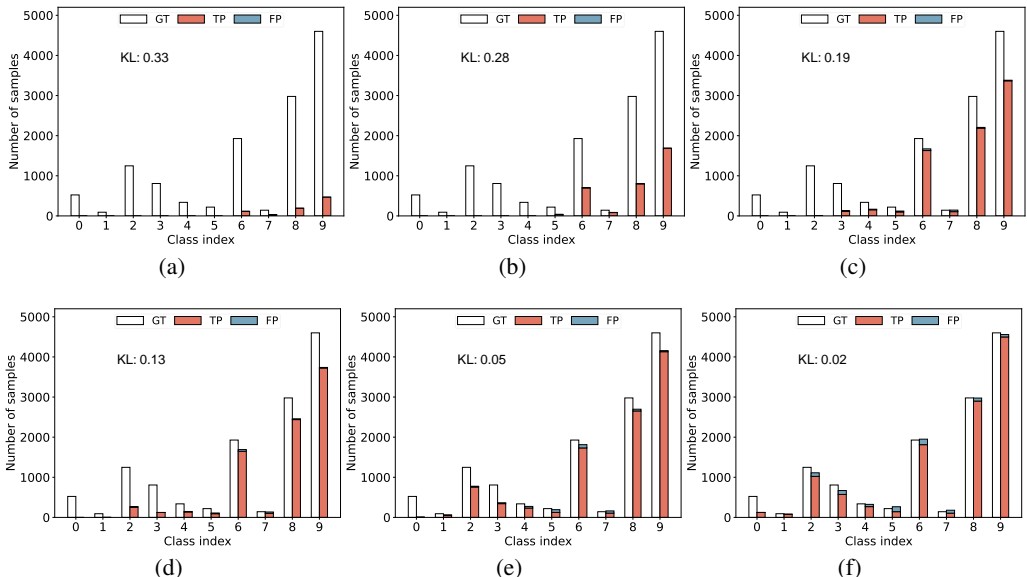

Figure 5: Evolution of pseudo-label predictions of our method under arbitrary unlabeled data distribution. GT denotes the ground-truth unlabeled data distribution. TP (FP) denotes the predicted true (false) positive pseudo-labels. KL denotes the Kullback-Leibler divergence between the predicted and ground-truth unlabeled data distributions. The dataset is CIFAR-10-LT with $(N_{max}, M_{max}, \gamma_l, \gamma_u) = (400, 4600, 50, 50)$. Our CPG can progressively increase the utilization rate of pseudo-labels while maintaining the high accuracy during training, as the pseudo-label distribution gradually approximates the ground-truth unlabeled data distribution.

# B Proof of Theorem 1

We first copy the Theorem 1 here.

**Theorem 1.** *Suppose that the loss function $\ell_{la}(f(x), y)$ is $\rho$-Lipschitz with respect to $f(x)$ for all $y \in \{1, \ldots, C\}$ and upper-bounded by $U$. Given the pseudo-labeling error rate $0 < \epsilon < 1$, for any $\upsilon > 0$, with probability at least $1 - \upsilon$, we have:*

$$R_T \leq R_0 - \sum_{t=1}^{T} \lambda_t + U \sum_{t=1}^{T} \epsilon_t + 4\sqrt{2}\rho \sum_{t=1}^{T} \sum_{y=1}^{C} \mathcal{R}_{O_t}(\mathcal{H}_y) + 2U \sum_{t=1}^{T} \sqrt{\frac{\log \frac{2}{\upsilon}}{2O_t}}, \tag{11}$$

*where $T$ denotes the total number of training steps, $R_T$ and $R_0$ represent the model empirical risk at the final training step and initial training step (without pseudo-labels), respectively, and $\epsilon_t$ represents the pseudo-labeling error rate at the $t$-th training step.*

*Proof.* We first restate the definition of the model empirical risk $R$ at the $t$-th training step, i.e.,

$$R_t = R_{t-1} - \lambda_t + I_{\epsilon_t} + I_{O_t}, \tag{12}$$

where $R_t$ denotes the model empirical risk at the $t$-th training step, $\lambda_t \geq 0$ quantifies the empirical risk reduction achieved, and $I_{\epsilon_t}$ and $I_{O_t}$ represent the impact of noisy pseudo-labels and number of training samples, respectively. We derive $I_{\epsilon_t}$ and $I_{O_t}$ below, omitting the subscript $t$ for brevity.

We define the true risk on the training dataset with respect to the classification model $f(x; \theta)$ as:

$$R(f) = \mathbb{E}_{(x,y)} \left[ \ell_{la}(f(x), y) \right]. \tag{13}$$

We aim to learn a good classification model by minimizing the empirical risk $\widehat{R}(f) = \widehat{R}_l(f) + \widehat{R}_u(f)$, with $\widehat{R}_l(f) = \frac{1}{N} \sum_{i=1}^{N} \ell_{la}(f(x_i), y_i)$ and $\widehat{R}_u(f) = \frac{1}{M} \sum_{j=1}^{M} \ell_{la}(f(x_j), y_j)$ representing the empirical risk on labeled and unlabeled training dataset, respectively.

Next, we derive the uniform deviation bound $I_O$ between the true risk $R(f)$ and the empirical risk $\widehat{R}(f)$ using the following lemma.

**Lemma 1.** *Suppose that the loss function $\ell_{la}(f(x), y)$ is $\rho$-Lipschitz with respect to $f(x)$ for all $y \in \{1, \ldots, C\}$ and upper-bounded by $U$. For any $\upsilon > 0$, with probability at least $1 - \upsilon$, we have:*

$$|R(f) - \widehat{R}(f)| \leq 2\sqrt{2}\rho \sum_{y=1}^{C} \mathcal{R}_{N+M}(\mathcal{H}_y) + U\sqrt{\frac{\log \frac{2}{\upsilon}}{2(N + M)}}, \tag{14}$$

*where the function space $\mathcal{H}_y$ for the label $y \in \{1, \ldots, C\}$ is $\{h : x \longmapsto f_y(x) | f \in \mathcal{F}\}$.*

*Proof.* In order to prove this lemma, we define the Rademacher complexity of the composition of loss function $\ell_{la}$ and model $f \in \mathcal{F}$ with $N$ labeled and $M$ unlabeled training samples as follows:

$$
\begin{aligned}
&\mathcal{R}_{N+M}(\ell_{la} \circ \mathcal{F}) \\
=&\mathbb{E}_{(x,y,\mu)} \left[ \sup_{f \in \mathcal{F}} \sum_{i=1}^{N} \mu_i \left( \ell_{la}(f(x_i), y_i) \right) + \sum_{j=1}^{M} \mu_j \left( \ell_{la}(f(x_j), y_j) \right) \right] \\
\leq& \sqrt{2}\rho \sum_{y=1}^{C} \mathcal{R}_{N+M}(\mathcal{H}_y),
\end{aligned}
\tag{15}
$$

where $\circ$ denotes the function composition operator, $\mathbb{E}_{(x,y,\mu)}$ denotes the expectation over $x$, $y$, and $\mu$, $\mu$ denotes the Rademacher variable, $\sup_{f \in \mathcal{F}}$ denotes the supremum (or least upper bound) over the function set $\mathcal{F}$ of model $f$. The second line holds because of the Rademacher vector contraction inequality [43].

Then, we proceed with the proof by showing that the one direction $\sup_{f \in \mathcal{F}} R(f) - \widehat{R}(f)$ is bounded with probability at least $1 - \upsilon/2$, and the other direction $\sup_{f \in \mathcal{F}} \widehat{R}(f) - R(f)$ can be proved similarly. Note that replacing a sample $(x_i, y_i)$ leads to a change of $\sup_{f \in \mathcal{F}} R(f) - \widehat{R}(f)$ at most $\frac{U}{N+M}$ due to the fact that $\ell_{la}$ is bounded by $U$. According to the McDiarmid's inequality [38], for any $\upsilon > 0$, with probability at least $1 - \upsilon/2$, we have:

$$\sup_{f \in \mathcal{F}} R(f) - \widehat{R}(f) \leq \mathbb{E}\left[ \sup_{f \in \mathcal{F}} R(f) - \widehat{R}(f) \right] + U\sqrt{\frac{\log \frac{2}{\upsilon}}{2(N + M)}}. \tag{16}$$

According to the result in [38] that shows $\mathbb{E}\left[ \sup_{f \in \mathcal{F}} R(f) - \widehat{R}(f) \right] \leq 2\mathcal{R}_{N+M}(\mathcal{F})$, and further considering the other direction $\sup_{f \in \mathcal{F}} \widehat{R}(f) - R(f)$, with probability at least $1 - \upsilon$, we have:

$$\sup_{f \in \mathcal{F}} |R(f) - \widehat{R}(f)| \leq 2\sqrt{2}\rho \sum_{y=1}^{C} \mathcal{R}_{N+M}(\mathcal{H}_y) + U\sqrt{\frac{\log \frac{2}{\upsilon}}{2(N + M)}}, \tag{17}$$

which completes the proof.

$\square$

In SSL, we cannot minimize the empirical risk $\widehat{R}_u(f)$ directly, since the ground-truth labels are inaccessible for unlabeled training data. Therefore, we need to train the model with $\widehat{R}'_u(f) =$

$\frac{1}{\hat{M}} \sum_{j=1}^{\hat{M}} \ell_{la}(f(x_j), \hat{y}_j)$, where $\hat{y}_j$ denotes the pseudo-label of unlabeled sample $x_j$, and $\hat{M}$ quantifies the pseudo-labels identified by the dynamic controllable filtering. Let $\hat{f} = \arg\min_{f \in \mathcal{F}} \widehat{R}(f)$ be the empirical risk minimizer, and $f^* = \arg\min_{f \in \mathcal{F}} R(f)$ be the true risk minimizer. Let $\ell_{ours} = \frac{1}{N+\hat{M}} \left[ \sum_{i=1}^{N} \ell_{la}(f(x_i), y_i) + \sum_{j=1}^{\hat{M}} \ell_{la}(f(x_j), \hat{y}_j) \right]$ be the training loss on $N$ labeled and $\hat{M}$ pseudo-labeled samples with pseudo-labeling error rate $\epsilon$ for our method.

Then, we can bound the difference $I_{\epsilon_t}$ between $\widehat{R}_u(f)$ and $\widehat{R}'_u(f)$ as follows.

**Lemma 2.** *Suppose that the loss function $\ell_{la}(f(x), y)$ is $\rho$-Lipschitz with respect to $f(x)$ for all $y \in \{1, \dots, C\}$ and upper-bounded by $U$. Given the pseudo-labeling error rate $0 < \epsilon < 1$, we have:*

$$0 \le |\widehat{R}'_u(f) - \widehat{R}_u(f)| \le U\epsilon. \tag{18}$$

*Proof.* Let's first expand $\widehat{R}'_u(f)$:

$$
\begin{aligned}
\widehat{R}'_u(f) &= \frac{1}{\hat{M}} \sum_{j=1}^{\hat{M}} \ell_{la}(f(x_j), \hat{y}_j) \\
&= \frac{1}{\hat{M}} \sum_{j=1}^{\hat{M}} \left[ (1-\epsilon)\ell_{la}(f(x_j), y_j) + \frac{\epsilon}{C-1} \sum_{c=1}^{C} \mathbb{I}(\hat{y}_j \ne y_j)\ell_{la}(f(x_j), \hat{y}_j) \right].
\end{aligned}
\tag{19}
$$

Next, we show its upper bound:

$$
\begin{aligned}
\widehat{R}'_u(f) &\le \widehat{R}_u(f) + \frac{1}{\hat{M}} \sum_{j=1}^{\hat{M}} \left[ \frac{\epsilon}{C-1} \sum_{c=1}^{C} \mathbb{I}(\hat{y}_j \ne y_j)\ell_{la}(f(x_j), \hat{y}_j) \right] \\
&\le \widehat{R}_u(f) + \frac{1}{\hat{M}} \sum_{j=1}^{\hat{M}} \left[ \frac{\epsilon}{C-1} \sum_{c=1}^{C} \mathbb{I}(\hat{y}_j \ne y_j)U \right] \\
&\le \widehat{R}_u(f) + U\epsilon,
\end{aligned}
\tag{20}
$$

and when $\epsilon \to 0$, it reaches the lower bound $\widehat{R}_u(f)$, which concludes the proof.

$\square$

Based on the above lemmas, for any $\upsilon > 0$, with probability at least $1 - \upsilon$, at the $t$-th training step, we have:

$$
\begin{aligned}
R_t(\hat{f}) &\le \widehat{R}_t(\hat{f}) + 2\sqrt{2}\rho \sum_{y=1}^{C} \mathcal{R}_{O_t}(\mathcal{H}_y) + U\sqrt{\frac{\log \frac{2}{\upsilon}}{2O_t}} \\
&\le \widehat{R}_{t_l}(\hat{f}) + \widehat{R}_{t_u}(\hat{f}) + 2\sqrt{2}\rho \sum_{y=1}^{C} \mathcal{R}_{O_t}(\mathcal{H}_y) + U\sqrt{\frac{\log \frac{2}{\upsilon}}{2O_t}} \\
&\le \widehat{R}_{t_l}(\hat{f}) + \widehat{R}'_{t_u}(\hat{f}) + 2\sqrt{2}\rho \sum_{y=1}^{C} \mathcal{R}_{O_t}(\mathcal{H}_y) + U\sqrt{\frac{\log \frac{2}{\upsilon}}{2O_t}} \\
&\le \widehat{R}_{t_l}(f) + \widehat{R}'_{t_u}(f) + 2\sqrt{2}\rho \sum_{y=1}^{C} \mathcal{R}_{O_t}(\mathcal{H}_y) + U\sqrt{\frac{\log \frac{2}{\upsilon}}{2O_t}} \\
&\le \widehat{R}_{t_l}(f) + \widehat{R}_{t_u}(f) + U\epsilon_t + 2\sqrt{2}\rho \sum_{y=1}^{C} \mathcal{R}_{O_t}(\mathcal{H}_y) + U\sqrt{\frac{\log \frac{2}{\upsilon}}{2O_t}} \\
&\le \widehat{R}_t(f) + U\epsilon_t + 2\sqrt{2}\rho \sum_{y=1}^{C} \mathcal{R}_{O_t}(\mathcal{H}_y) + U\sqrt{\frac{\log \frac{2}{\upsilon}}{2O_t}}
\end{aligned}
$$

$$\leq R_t(f) + U\epsilon_t + 4\sqrt{2}\rho \sum_{y=1}^{C} \mathcal{R}_{O_t}(\mathcal{H}_y) + 2U\sqrt{\frac{\log \frac{2}{v}}{2O_t}}, \tag{21}$$

where the first and seventh lines are based on Lemma 1, and three and fifth lines are due to Lemma 2. The fourth line is by the definition of $\hat{f}$.

Then, we can derive $I_{\epsilon_t} + I_{O_t}$ as follows:

$$I_{\epsilon_t} + I_{O_t} = |\widehat{R}'_t(f) - \widehat{R}_t(f)| + |R_t(f) - \widehat{R}_t(f)| = |\widehat{R}'_t(f) - R_t(f)|$$

$$\leq U\epsilon_t + 4\sqrt{2}\rho \sum_{y=1}^{C} \mathcal{R}_{O_t}(\mathcal{H}_y) + 2U\sqrt{\frac{\log \frac{2}{v}}{2O_t}}, \tag{22}$$

Substituting Eq. (22) into Eq. (12) yields:

$$
\begin{cases}
R_t \leq R_{t-1} - \lambda_t + U\epsilon_t + 4\sqrt{2}\rho \sum_{y=1}^{C} \mathcal{R}_{O_t}(\mathcal{H}_y) + 2U\sqrt{\frac{\log \frac{2}{v}}{2O_t}}, \\[2mm]
R_{t-1} \leq R_{t-2} - \lambda_{t-1} + U\epsilon_{t-1} + 4\sqrt{2}\rho \sum_{y=1}^{C} \mathcal{R}_{O_{t-1}}(\mathcal{H}_y) + 2U\sqrt{\frac{\log \frac{2}{v}}{2O_{t-1}}}, \\[2mm]
\quad \vdots \\[2mm]
R_1 \leq R_0 - \lambda_1 + U\epsilon_1 + 4\sqrt{2}\rho \sum_{y=1}^{C} \mathcal{R}_{O_1}(\mathcal{H}_y) + 2U\sqrt{\frac{\log \frac{2}{v}}{2O_1}}.
\end{cases} \tag{23}
$$

Finally, combining the sub-equations in Eq. (23) through summation yields:

$$R_T \leq R_0 - \sum_{t=1}^{T} \lambda_t + U\sum_{t=1}^{T} \epsilon_t + 4\sqrt{2}\rho \sum_{t=1}^{T}\sum_{y=1}^{C} \mathcal{R}_{O_t}(\mathcal{H}_y) + 2U\sum_{t=1}^{T}\sqrt{\frac{\log \frac{2}{v}}{2O_t}}. \tag{24}$$

At this point, we have proven Theorem 1.

$\square$

## C  Dataset Details

We perform our experiments on four widely-used datasets (CIFAR-10-LT [39], CIFAR-100-LT [39], Food-101-LT [40], and SVHN-LT [41]), following the main experimental settings in FreeMatch [8] and SimPro [13], details are as below.

- *CIFAR-10-LT*: We test nine settings with $(N_{max}, M_{max}) = (400, 4600)$ and $\gamma \in \{100, 150, 200\}$. The unlabeled data follows either the same long-tailed distribution as the labeled data (consistent case), or an inverse long-tailed or arbitrary distribution (inconsistent cases).
- *CIFAR-100-LT*: We examine nine settings with $(N_{max}, M_{max}) = (50, 450)$ and $\gamma \in \{10, 15, 20\}$, using the same consistent/inconsistent unlabeled data distribution patterns as above.
- *Food-101-LT*: We evaluate six settings with $(N_{max}, M_{max}) = (50, 450)$ and $\gamma \in \{10, 15\}$, maintaining the same distribution cases.
- *SVHN-LT*: We evaluate three settings with $(N_{max}, M_{max}, \gamma) = (50, 450, 100)$, following the same distribution scheme.

## D  Implementation Details

We follow the default settings and hyperparameters in USB, i.e., the batch size of labeled data $B_l$ is set to 64, while unlabeled data $B_u$ is set to 7 times $B_l$, and the confidence threshold $\tau$ is set

to 0.95. Moreover, we use the WRN-28-2 [44] architecture, the SGD optimizer with momentum 0.9, and weight decay 5e-4 for training. We use a cosine learning rate decay [45] scheme, which sets the learning rate to $\eta \cos\left(\frac{7\pi t}{16T}\right)$, where the initial learning rate $\eta$ is set to 0.03, $t$ is the current training step, and the total number of training steps $T$ is set to $2^{18}$. We repeat each experiment with three different random seeds (i.e., 0, 1, and 2) and report the mean and standard deviation of the performances. We conduct the experiments on a single NVIDIA RTX 4090 GPU using PyTorch v2.3.1.

# E    More Experiment Results and Analyses

## E.1    Comparison of Macro-F1 on CIFAR-10-LT and CIFAR-100-LT

To better evaluate our method's performance under class imbalance, we include the Macro-F1 metric in our analysis. Table 7 presents the results on CIFAR-10-LT and CIFAR-100-LT. In both cases, the labeled data follows a long-tailed distribution, while the unlabeled data varies across long-tailed, inverse long-tailed, and arbitrary distributions. The results in Table 7 show that our method achieves significant

Table 7: Comparison of Macro-F1 (%) on CIFAR-10-LT ($N_{max} = 400, M_{max} = 4600, \gamma = 100$) and CIFAR-100-LT ($N_{max} = 50, M_{max} = 450, \gamma = 10$) under different unlabeled data distributions.

| Scenario | Method | CIFAR-10-LT | | | Avg. | CIFAR-100-LT | | | Avg. |
|---|---|---|---|---|---|---|---|---|---|
| | | $\gamma = 100$ | | | | $\gamma = 10$ | | | |
| | | Arbitrary | Inverse | Consistent | | Arbitrary | Inverse | Consistent | |
| SSL | FixMatch [7] | 53.52 | 54.24 | 66.47 | 58.07 | 44.58 | 44.85 | 46.46 | 45.30 |
| | FreetMatch [8] | 66.00 | 67.00 | 68.04 | 67.01 | 44.13 | 43.57 | 43.84 | 43.85 |
| | SoftMatch [9] | 62.49 | 66.66 | 71.53 | 66.89 | 45.67 | 46.73 | 46.91 | 46.44 |
| ReaLTSSL | ACR† [14] | 59.01 | 64.03 | 70.25 | 64.43 | 45.69 | 48.89 | 41.92 | 45.50 |
| | SimPro [13] | 61.22 | 62.11 | 57.81 | 60.38 | 39.98 | 41.54 | 41.87 | 41.13 |
| | CDMAD [42] | 61.73 | 62.88 | 62.11 | 62.24 | 32.97 | 35.80 | 36.68 | 35.15 |
| | Ours | **81.99** | **82.31** | **78.25** | **80.85** | **51.31** | **51.25** | **50.41** | **50.99** |

improvements in Macro-F1 over existing baselines, further validating its effectiveness for imbalance learning. Notably, it yields average Macro-F1 gains of **13.84 pp** on CIFAR-10-LT and 4.55 pp on CIFAR-100-LT.

## E.2    Evaluation on Audio Modality

While our method is evaluated on vision tasks, the core principles of CPG can be extended to other modalities. Here, we further evaluate our method on the audio modality. Specifically, for audio data, this adaptation requires two key modifications: (i) replacing vision-specific backbones (e.g., WRN-28-2 [44]) with audio-compatible models (e.g., HuBERT [46]), and (ii) substituting image augmentations with audio-specific ones. To demonstrate this adaptability, we perform our CPG on ESC-50-LT [47], a standard benchmark for environmental sound classification. As demonstrated in Table 8, our method achieves state-of-the-art performance on this audio benchmark, confirming its effec-

Table 8: Comparison of accuracy (%) on ESC-50-LT ($N_{max} = 12, M_{max} = 12, \gamma = 1.2$) under long-tailed labeled data and arbitrary unlabeled data distributions.

| FreeMatch [8] | SimPro [13] | Ours |
|---|---|---|
| 68.13 | 68.75 | **69.13** |

tiveness beyond the visual modality. These results strongly support the generalizability of CPG's framework across different modalities.

## E.3    Evaluation under Noisy Scenarios

To comprehensively evaluate our method's robustness under noisy scenarios, we conduct additional experiments on CIFAR-10-LT with ($N_{max} = 400, M_{max} = 4600, \gamma = 100$), under varying noise rates (i.e., 0%, 10%, 20%, and 30%). The labeled data follows a long-tailed distribution, while the unlabeled data adheres to an arbitrary distribution. As demonstrated in Table 9, our method consistently outperforms baselines in noisy settings,

Table 9: Comparison of accuracy (%) on CIFAR-10-LT ($N_{max} = 400, M_{max} = 4600, \gamma = 100$) across different noise rate scenarios.

| Noise rate | 0% | 10% | 20% | 30% | $\Delta$ (0% → 30%) |
|---|---|---|---|---|---|
| FreetMatch [8] | 66.33 | 60.33 | 56.24 | 51.29 | -15.04 |
| SimPro [13] | 64.73 | 59.86 | 44.54 | 34.80 | -29.93 |
| Ours | **82.33** | **75.95** | **71.82** | **69.87** | **-12.46** |

achieving the highest accuracy across all noise rates with minimal performance degradation. Notably, when noise rate increases from 0% to 30%, our method shows a degradation ($\Delta$) of only 12.46

pp, compared to 15.04 pp for FreeMatch [8] and 29.93 pp for SimPro [13]. This demonstrates our method's strong robustness to label noise.

### E.4 Evaluation on Large-scale and Realistic Datasets

The Food-101 dataset (evaluated in Table 3) is a widely adopted benchmark for food image classification. Its training set naturally contains a proportion of label noise, making the data distribution implicitly unknown. Thus, distribution mismatches between labeled and unlabeled data occur when splitting the training set into labeled and unlabeled datasets. Despite these challenges, our method achieves a significant average accuracy gain of 2.57 pp on this dataset.

Table 10: Comparison of accuracy (%) on ImageNet-127.

| Scenario | Method | 32×32 | 64×64 |
|---|---|---|---|
| SSL | FixMatch [7] | 29.57 | 37.40 |
| | FreetMatch [8] | 31.37 | 39.67 |
| | SoftMatch [9] | 31.50 | 39.40 |
| ReaLTSSL | ACR† [14] | 38.47 | 47.61 |
| | SimPro [13] | 43.31 | 46.93 |
| | CDMAD [42] | 15.46 | 21.92 |
| | Ours | **44.58** | **50.43** |

To further evaluate our method's effectiveness in large-scale and real-world scenarios, we conduct experiments on ImageNet-127 (an imbalanced dataset, 127 classes, imbalance ratio 286) with an arbitrary (non-long-tailed) distribution. As shown in Table 10, our method consistently outperforms baselines, achieving gains of 1.27 pp at 32×32 resolution and 2.82 pp at 64×64 resolution. These results highlight our method's robustness in large-scale and realistic settings beyond standard benchmarks.

### E.5 Evaluation under Extreme Settings

In long-tailed learning scenarios, there exists an inherent trade-off between extremely small $N_{max}$ (number of samples in the most frequent class) and large imbalance ratio $\gamma$ values, since $N_{min} = N_{max}/\gamma \geq 1$. Our initial experiments on CIFAR-10-LT in Table 1 already employed challenging settings ($N_{max} = 400, M_{max} = 4600, \gamma = 100$) with $N_{min} = 4$. Notably, even with such extremely few tail class samples, our method still demonstrates significant performance gains over baselines.

Table 11: Comparison of accuracy (%) on CIFAR-10-LT under extreme scenarios.

| Method | Setting A1 | Setting A2 | Setting B |
|---|---|---|---|
| FreetMatch [8] | 63.86 | 45.75 | 55.92 |
| SimPro [13] | 31.01 | 15.20 | 50.98 |
| Ours | **69.56** | **49.28** | **77.78** |

To further evaluate our method, we conduct additional experiments under three more extreme settings: extreme sparsity setting A1 ($N_{max} = 100, N_{min} = 1, M_{max} = 4900, \gamma = 100$), extreme sparsity setting A2 ($N_{max} = 10, N_{min} = 1, M_{max} = 4990, \gamma_l = 10, \gamma_u = 300$), and extreme imbalance setting B ($N_{max} = 400, N_{min} = 1, M_{max} = 4600, \gamma = 300$). As demonstrated in Table 11, our method achieves state-of-the-art performance across three settings, attaining 69.56 pp in setting A1, 49.28 pp in setting A2, and 77.78 pp in setting B. These results represent significant improvements over all baseline methods, underscoring our method's exceptional capability to handle extreme settings.

### E.6 Evaluation using Other Architectures

To evaluate architectural generalization, we conduct experiments using ResNet-50 on CIFAR-10-LT with ($N_{max} = 400, M_{max} = 4600, \gamma = 100$). The labeled data follows a long-tailed distribution, while the unlabeled data conforms to either an inverse long-tailed or an arbitrary distribution. As evidenced by Table 12, our method demonstrates significant improvements, achieving absolute performance gains of 7.85 pp and 7.23 pp over current state-of-the-art methods in these respective scenarios. These results highlight our method's consistent robustness when applied to different network architectures and its ability to handle varying distribution patterns in the unlabeled data.

Table 12: Comparison of accuracy (%) on CIFAR-10-LT using ResNet-50.

| Method | Arbitrary | Inverse |
|---|---|---|
| FreetMatch [8] | 47.68 | 48.28 |
| SimPro [13] | 43.67 | 51.26 |
| Ours | **55.53** | **58.49** |

## F More Ablation Studies and Analyses

### F.1 Ablation Study on the Confidence Threshold

The confidence threshold $\tau = 0.95$ is a common setting in SSL, which we adopt for a fair comparison. To evaluate the impact of different confidence thresholds on model performance, we conduct

additional experiments on CIFAR-10-LT with ($N_{max} = 400, M_{max} = 4600, \gamma = 100$) and $\tau \in \{0.75, 0.85, 0.95\}$. As shown in Table 13, our method's performance variations across different thresholds remain relatively small (within 1-2 pp), and it maintains consistently superior performance compared to baseline methods. This threshold-insensitive property highlights the stability of our method.

Table 13: Ablation study on the confidence threshold.

| Method | Arbitrary | Inverse | Consistent |
|---|---|---|---|
| FreetMatch [8] | 66.33 | 68.10 | 69.42 |
| SimPro [13] | 64.73 | 65.25 | 64.37 |
| Ours ($\tau = 0.75$) | 81.35 | 78.26 | 76.20 |
| Ours ($\tau = 0.85$) | 81.89 | 80.85 | 77.79 |
| **Ours ($\tau = 0.95$)** | **82.33** | **82.32** | **78.35** |

### F.2 Ablation Study on the Auxiliary Branch

To evaluate the impact of the auxiliary branch, we integrate it into both FreeMatch [8] and SimPro [13] and analyze their performance variations. As demonstrated in Figs. 6 and 7, our analysis reveals that the auxiliary branch exhibits divergent effects across methods. FreeMatch‡ (FreeMatch with the auxiliary branch, Fig. 6(a)) selectively improves pseudo-label quality and quantity for both majority classes (e.g., classes 8 and 9) and minority classes (e.g., class 7), unlike the baseline (FreeMatch without the auxiliary branch, Fig. 1(a)). In contrast, SimPro suffers consistent degradation in pseudo-label quality and quantity across nearly all classes when integrated with the auxiliary branch. These trends are further supported by the error rate and utilization rate comparisons in Figs. 3 and 7, which indicate similar performance variations.

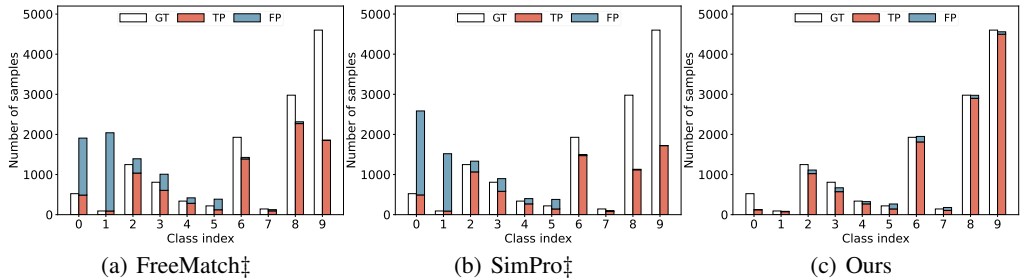

(a) FreeMatch‡          (b) SimPro‡          (c) Ours

Figure 6: Comparison of pseudo-label predictions among FreeMatch‡ [8], SimPro‡ [13], and our CPG under arbitrary unlabeled data distribution. GT denotes the ground-truth unlabeled data distribution. TP (FP) denotes the predicted true (false) positive pseudo-labels. The dataset is CIFAR-10-LT with $(N_{max}, M_{max}, \gamma_l, \gamma_u) = (400, 4600, 50, 50)$. Our CPG can generate more reliable pseudo-labels than FreeMatch‡ and SimPro‡ in both minority classes like class 1, 2 and majority classes like class 6, 8, 9. ‡ indicates that we reproduce baseline methods with the auxiliary branch, similar to our CPG.

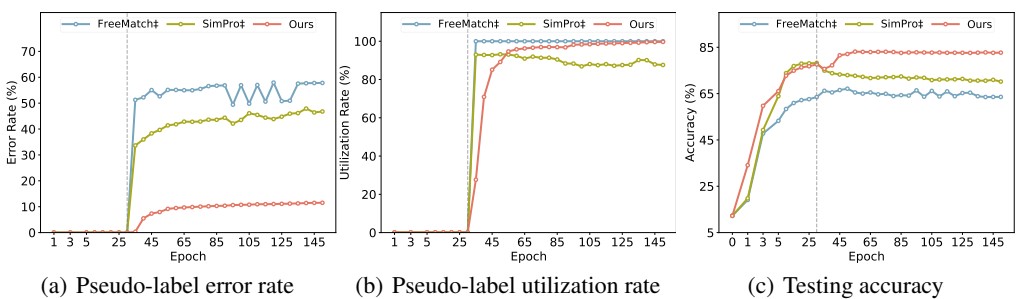

(a) Pseudo-label error rate     (b) Pseudo-label utilization rate     (c) Testing accuracy

Figure 7: Comparison of pseudo-label error rate (a), pseudo-label utilization rate (b), and testing accuracy (c) among FreeMatch‡ [8], SimPro‡ [13], and our CPG under arbitrary unlabeled data distribution. The dataset is CIFAR-10-LT with $(N_{max}, M_{max}, \gamma_l, \gamma_u) = (400, 4600, 50, 50)$. The vertical gray dotted line indicates the initiation of pseudo-labeling. Our CPG can generate pseudo-labels with a lower error rate and comparable utilization rate, achieving superior testing accuracy compared to both FreeMatch‡ and SimPro‡. ‡ indicates that we reproduce baseline methods with the auxiliary branch, similar to our CPG.

### F.3 Ablation Study on the Anchor Distributions

In the main paper, we compare our CPG with ACR† (without anchor distributions) since anchor distributions are typically unknown in real-world scenarios. Here, we further supplement this analysis by comparing CPG with ACR (with anchor distributions) under the assumption that anchor distributions are known. As shown in Tables 14, 15, and 16, our method outperforms ACR even when anchor distributions are known, while ACR relies on this prior knowledge. This demonstrates the robustness and practical applicability of our method.

Table 14: Comparison of accuracy (%) on CIFAR-10-LT ($N_{max} = 400, M_{max} = 4600$) with class imbalance ratio $\gamma \in \{100, 150, 200\}$ under different unlabeled data distributions.

| Method | $\gamma = 100$ | | | $\gamma = 150$ | | | $\gamma = 200$ | | | Avg. |
|---|---|---|---|---|---|---|---|---|---|---|
| | Arbitrary | Inverse | Consistent | Arbitrary | Inverse | Consistent | Arbitrary | Inverse | Consistent | |
| ACR† [14] | 60.60 ±4.21 | 62.54 ±4.04 | 73.20 ±1.80 | 48.22 ±8.07 | 51.08 ±1.03 | 68.31 ±0.35 | 50.13 ±7.50 | 53.40 ±3.49 | 65.12 ±1.05 | 59.18 ±3.51 |
| ACR [14] | 76.18 ±0.91 | 75.72 ±1.05 | 73.27 ±0.96 | 66.69 ±2.03 | 66.68 ±1.82 | 69.34 ±3.05 | 68.91 ±3.50 | 67.25 ±1.57 | 66.48 ±4.79 | 70.06 ±2.19 |
| Ours | **82.10** ±0.74 | **82.37** ±0.15 | **76.93** ±2.68 | **76.38** ±3.87 | **78.19** ±1.63 | **70.75** ±2.13 | **75.18** ±3.28 | **77.45** ±1.44 | **68.33** ±3.94 | **76.41** ±2.21 |

Table 15: Comparison of accuracy (%) on CIFAR-100-LT ($N_{max} = 50, M_{max} = 450$) with class imbalance ratio $\gamma \in \{10, 15, 20\}$ under different unlabeled data distributions.

| Method | $\gamma = 10$ | | | $\gamma = 15$ | | | $\gamma = 20$ | | | Avg. |
|---|---|---|---|---|---|---|---|---|---|---|
| | Arbitrary | Inverse | Consistent | Arbitrary | Inverse | Consistent | Arbitrary | Inverse | Consistent | |
| ACR† [14] | 44.10 ±1.59 | 48.50 ±0.86 | 41.33 ±0.41 | 35.12 ±1.36 | 39.01 ±0.38 | 32.73 ±1.74 | 29.08 ±1.11 | 32.78 ±1.59 | 29.69 ±2.94 | 36.93 ±1.33 |
| ACR [14] | 50.92 ±0.96 | 50.29 ±0.71 | 50.57 ±0.98 | 44.02 ±0.63 | 43.49 ±0.54 | 44.68 ±0.90 | 41.93 ±1.64 | 39.62 ±1.53 | 42.28 ±1.29 | 45.31 ±1.02 |
| Ours | **51.48** ±0.32 | **51.46** ±0.22 | **51.22** ±0.71 | **46.26** ±1.40 | **47.88** ±0.24 | **45.94** ±1.02 | **44.17** ±1.30 | **44.17** ±1.01 | **42.66** ±0.56 | **47.25** ±0.75 |

Table 16: Comparison of accuracy (%) on Food-101-LT ($N_{max} = 50, M_{max} = 450$) with class imbalance ratio $\gamma \in \{10, 15\}$ and SVHN-LT ($N_{max} = 400, M_{max} = 4600, \gamma = 100$) under different unlabeled data distributions.

| Method | Food-101-LT | | | | | | Avg. | SVHN-LT | | | Avg. |
|---|---|---|---|---|---|---|---|---|---|---|---|
| | $\gamma = 10$ | | | $\gamma = 15$ | | | | $\gamma = 100$ | | | |
| | Arbitrary | Inverse | Consistent | Arbitrary | Inverse | Consistent | | Arbitrary | Inverse | Consistent | |
| ACR† [14] | 18.41 ±0.75 | 19.27 ±1.12 | 17.30 ±0.58 | 15.44 ±0.50 | 17.07 ±0.61 | 13.88 ±0.73 | 16.90 ±0.72 | 87.67 ±4.60 | 84.20 ±1.58 | 92.64 ±0.50 | 88.17 ±2.23 |
| ACR [13] | 21.87 ±1.24 | 22.33 ±0.10 | 21.81 ±0.47 | 19.11 ±1.49 | 19.04 ±0.56 | 18.84 ±0.13 | 20.50 ±0.67 | 88.04 ±1.70 | 90.04 ±0.58 | 89.59 ±1.54 | 89.22 ±1.28 |
| Ours | **25.98** ±0.66 | **25.52** ±0.43 | **25.24** ±0.30 | **21.88** ±0.85 | **21.27** ±0.51 | **22.11** ±0.72 | **23.67** ±0.58 | **93.99** ±0.34 | **94.74** ±0.49 | **93.45** ±0.52 | **94.06** ±0.45 |

## G Relation to Existing Methods

Here, we summarize the difference between our CPG and existing methods (i.e., FixMatch [7], FlexMatch [29], UPS [48], CADR [49], FlexDA [50], DASO [51]).

While CPG adopts confidence-based filtering similar to FixMatch [7] and FlexMatch [29], its key innovation is the controllable self-reinforcing optimization cycle, which incorporates reliable pseudo-labels from the unlabeled dataset into the labeled dataset rather than employing consistency regularization, distinguishing it from existing methods. This cycle operates in three steps: (i) expanding the labeled dataset with reliable pseudo-labels, (ii) constructing a Bayes-optimal classifier, and (iii) iteratively improving pseudo-label quality, forming a theoretically grounded feedback loop. Additionally, CPG ensures full sample utilization via an auxiliary branch and reduces label noise accumulation through a voting-based stabilization technique, which enforces consistent pseudo-label assignments across training steps. As evidenced by the ablation studies in Table 5, these components individually contribute significant performance improvements, with the auxiliary branch and optimization cycle yielding average gains of 1.82 pp and 6.97 pp, respectively.

The goals of UPS [48] and CADR [49] differ from ours. UPS targets poor model calibration in SSL without distribution mismatch and CADR addresses distribution mismatch due to label missing not at random, while our work focuses on handling the unknown and arbitrary unlabeled data distribution. Specifically, UPS mitigates high-confidence incorrect pseudo-labels caused by poor model calibration through uncertainty estimation techniques like MC-Dropout [52], and CADR focuses on resolving classifier bias due to label missing not at random by employing class-aware propensity estimation and dynamic threshold adjustment for pseudo-label selection. Unlike these existing techniques that rely on uncertainty estimation or threshold adjustment, we propose a novel controllable self-reinforcing optimization cycle that operates without distribution estimation or correction, introducing distinct conceptual and methodological innovations. More importantly, our framework is specifically designed to address real-world scenarios with unknown and arbitrary unlabeled data distributions.

While FlexDA [50] relies on estimating the unlabeled data distribution for distribution alignment and pseudo-label generation, our core contribution introduces a controllable self-reinforcing optimization cycle that entirely bypasses unlabeled data distribution estimation. This approach is fundamentally different from FlexDA's dynamic distribution alignment. Specifically, we iteratively expand the labeled dataset with reliable pseudo-labels, train a Bayes-optimal classifier via logit adjustment on the updated labeled dataset without using the pseudo-label distribution. More importantly, we further theoretically prove that this optimization cycle can significantly reduce the generalization error, whereas FlexDA lacks such theoretical guarantees. Moreover, our auxiliary branch leverages all available samples (without a confidence threshold) to enhance feature learning while preventing the primary branch classifier from error pseudo-labels. Notably, prior methods (e.g., FlexDA [50], SimPro [50], UPS [48], DASO [51]) lack class-aware enhancements for tail classes.

## H  Limitations

(i) Our CPG assumes that the labeled data is free from noise during training. Consequently, CPG may encounter challenges when adapting to scenarios involving noisy labels. (ii) Our CPG operates under the assumption that labeled data follows a long-tailed distribution, with even tail classes retaining minimal supervision. However, its performance may degrade when labeled data follows a uniform distribution but suffers from extremely limited supervision. Future work will explore extensions to address these limitations and improve model generalization.

## I  Compute Resources

- CPU: AMD EPYC 7642 48-Core Processor $\times$ 2
- GPU: NVIDIA GeForce RTX 4090 24G $\times$ 1
- MEM: 500G
- Maximum total computing time: training + testing $\approx$ 17h

## J  Broader Impacts

This paper presents work whose goal is to advance the field of Machine Learning. Specifically, we propose a new realistic long-tailed semi-supervised learning algorithm to improve performance in unknown arbitrary distribution scenarios by expanding the labeled dataset with the progressively identified reliable pseudo-labels from the unlabeled dataset and training the model on the updated labeled dataset with a known distribution, making it unaffected by the unlabeled data distribution. We further theoretically prove that this iterative process can significantly reduce the generalization error under some conditions. There are many potential societal consequences of our work, none which we feel must be specifically highlighted here.

