# OpenReview forum: "Keep It on a Leash: Controllable Pseudo-label Generation Towards Realistic Long-Tailed Semi-Supervised Learning"
_NeurIPS.cc/2025/Conference — NeurIPS 2025 poster_

### Official Review · Reviewer_HaKG · 2025-06-30

**Clarity:** 3
**Significance:** 2
**Originality:** 2
**Rating:** 3
**Confidence:** 4

**Summary:**

This paper proposes a method for realistic long-tailed semi-supervised learning (ReaLTSSL), where the distributions of labeled and unlabeled data may differ. To address the challenge of unknown and potentially arbitrary unlabeled distributions, the authors propose to bypass direct distribution estimation. Instead, the method progressively identifies reliable pseudo-labels and adds them to the labeled set, forming an updated dataset with a known class distribution. A Bayes-optimal classifier is then trained using logit-adjusted loss based on this controlled distribution.

The core of the approach is the Controllable Self-reinforcing Optimization Cycle, which uses agreement between weak and strong augmentation predictions to select consistent pseudo-labels. These labels are treated as reliable and used to update the training distribution in a feedback loop. To further enhance performance, the method introduces a Class-aware Adaptive Augmentation (CAA) module that strengthens minority class representations based on intra-class compactness, and an Auxiliary Branch that enforces prediction consistency across augmentations to better utilize all samples.

The experimental evaluation is comprehensive. The authors test on four benchmark datasets: CIFAR-10-LT, CIFAR-100-LT, SVHN-LT, and Food-101-LT under a variety of distribution mismatch settings, including long-tailed, inverse, and arbitrary configurations. The method achieves consistent improvements over strong baselines in both aligned and mismatched label/unlabeled distribution settings.

**Questions:**

The proposed method starts training with logit adjustment applied to the limited and imbalanced labeled data, before incorporating any pseudo-labels. Under severe long-tailed settings, where the labeled sample size is extremely small, how reliable is this initial supervised classifier? Has the method been evaluated under more extreme label sparsity (e.g., lower N_max) to stress test this dependency?

How does the proposed method differ in principle from related realSLL works and other related works that also can handle distribution mismatch? can the authors clarify what is truly novel about the proposed pipeline beyond recombining existing techniques?

The authors introduce arbitrary unlabeled data distributions to simulate realistic challenges. However, are there real-world datasets or applications where such labeled/unlabeled distribution divergence naturally occurs?

The experiments use a single backbone (WideResNet-28-2). Has the method been tested on other architectures?

If these issues are carefully addressed, I would consider raising my score.

**Ethical Concerns:**

["NO or VERY MINOR ethics concerns only"]

**Final Justification:**

I acknowledge the authors' detailed rebuttal and appreciate the additional experiments addressing my earlier concerns, especially those related to performance under extreme settings and the comparison with FlexDA. However, after carefully reviewing all responses, my primary concerns regarding the conceptual novelty remain unresolved.

Specifically, the authors repeatedly emphasize that the novelty of their method is confirmed by other reviewers (e.g., TLMt, ydDN, Ckat), yet upon closely examining these reviewers' comments, Reviewer TLMt explicitly lists originality as a weakness rather than a strength.

The authors argue their novelty lies in the explicit integration of these methods into a "self-reinforcing optimization cycle." However, iterative training approaches involving pseudo-label updates are common in existing SSL frameworks (e.g., CReST’s iterative distribution adjustment). Label refinement cycles are inherent in many self-learning methods, although not explicitly introduced with the "self-reinforcing" terminology presented here. Thus, the claimed novelty is primarily terminological rather than genuinely methodological.

Furthermore, while the proposed method ("Leash") integrates established techniques, such as pseudo-label distribution correction, logit adjustment, consistency regularization, and class-aware strategies, these components have individually appeared frequently in prior literature. Moreover, the motivation closely aligns with the previously published method, "Flexible Distribution Alignment: Towards Long-tailed Semi-supervised Learning with Proper Calibration" (ECCV 2024), further diluting the originality of this submission. Finally, before the rebuttal, the experiments were restricted to a single backbone (WideResNet-28-2), and inconsistent class imbalance ratios were applied across different datasets. These experimental limitations weaken the generalizability and rigor required for NeurIPS acceptance.

While other reviewers have rated this paper highly, my independent evaluation suggests it does not reach the expected novelty and methodological standards required by NeurIPS. Therefore, I maintain my original rating (3: Borderline reject). However, I am OK whether this paper is accepted or not.

**Limitations:**

yes

**Paper Formatting Concerns:**

No major formatting issues

**Quality:**

2

**Strengths And Weaknesses:**

**Strength**

-The paper targets the mismatch between labeled and unlabeled data distributions,  which reflects real-world data collection scenarios. The proposed self-reinforcing cycle uses consistency between weak and strong views to select pseudo-labels and gradually improve the labeled distribution. This approach is simple to understand and avoids unstable threshold tuning.

 -The method is tested across four standard benchmarks (CIFAR-10/100-LT, SVHN-LT, Food-101-LT) and three types of distribution mismatches (consistent, inverse, arbitrary). The design of diverse imbalance ratios and distribution scenarios strengthens the generality of the results.

-The ablation studies isolate the contributions of each module (CAA, auxiliary branch, and optimization cycle), and the use of pairwise t-tests provides strong statistical support for performance claims.

**Weaknesses**

-The method combines several existing ideas, such as logit adjustment, augmentation-aware consistency, and pseudo-label voting, but **the conceptual novelty is relatively limited**.

-Similar goals have been addressed in earlier works like UPS [1] (uncertainty filtering based on model disagreement) and CADR[2] (reweighting pseudo-labels to correct distribution mismatch). These are not sufficiently discussed.

-The method still relies heavily on a **strong initial supervised phase**, and the assumption that the initial labeled set can provide stable logit-adjusted priors is not fully examined under extreme settings.


[1] IN DEFENSE OF PSEUDO-LABELING: AN UNCERTAINTY-AWARE PSEUDO-LABEL SELECTION FRAMEWORK FOR SEMI-SUPERVISED LEARNING

[2] ON NON-RANDOM MISSING LABELS IN SEMI-SUPERVISED LEARNING

---

> ### Author Rebuttal · Authors · 2025-07-31
>
> ### **The main novelty of the work (W1 & Q2)**
> Thanks for your comments. While our method leverages established components (e.g., logit adjustment, consistency regularization, and pseudo-label voting), its **core innovation is fundamentally different from prior works**.
>
> Existing methods typically employ high-confidence thresholds to filter pseudo-labels, then estimate unlabeled data distribution for subsequent refinement. However, during early training stages, the confidence threshold may completely exclude samples from certain classes. This leads to inaccurate estimation of the unlabeled data distribution, which in turn adversely affects subsequent pseudo-label generation steps that depend on this estimation. As shown in Figure 3, this leads to error accumulation and performance degradation.
>
> Our method introduces a controllable self-reinforcing optimization cycle that: (i) **progressively integrates reliable pseudo-labels from the unlabeled dataset into the labeled dataset**, and (ii) **trains on the updated labeled dataset, which follows a known distribution**. This strategy eliminates dependence on error-prone unlabeled data distribution estimation, effectively circumventing distribution estimation entirely. Meanwhile, **we theoretically prove that this optimization cycle can significantly reduce the generalization error**.
>
> Our method is unaffected by the unlabeled data distribution and addresses the unknown and arbitrary unlabeled data distribution, in contrast to prior works that attempt (and often fail) to estimate the unlabeled data distribution. As demonstrated in Table 5 of the paper, this cycle contributes an average performance improvement of **6.97%**, representing the most significant gain among all components in our framework. **Reviewers TLMt, ydDN, and Ckat all confirm our novelty**.
> ### **Discuss related work (W2)**
> The goals of UPS and CADR differ from ours. UPS targets poor model calibration in SSL without distribution mismatch and CADR addresses distribution mismatch due to label missing not at random, while our work focuses on handling the unknown and arbitrary unlabeled data distribution. Furthermore, our approach introduces distinct conceptual and methodological innovations. We provide a comprehensive comparison with both UPS and CADR below.
>
> UPS mitigates high-confidence incorrect pseudo-labels caused by poor model calibration through uncertainty estimation techniques like MC-Dropout. CADR focuses on resolving classifier bias due to label missing not at random by employing class-aware propensity estimation and dynamic threshold adjustment for pseudo-label selection. Unlike these existing techniques that rely on uncertainty estimation or threshold adjustment, we propose **a novel controllable self-reinforcing optimization cycle** that operates without distribution estimation or correction. More importantly, our framework is specifically designed to address real-world scenarios with unknown and arbitrary unlabeled data distributions. These discussion will be added in the final version.
> ### **Evaluation under extreme settings (W3 & Q1)**
> In long-tailed learning scenarios, there exists an inherent trade-off between extremely small $N_{max}$ (number of samples in the most frequent class) and large imbalance ratio $\gamma$ values, since $N_{min} = N_{max}/\gamma \geq 1$. Our initial experiments on CIFAR-10-LT in Table 1 already employed challenging settings ($N_{max}=400$, $M_{max}=4600$, $\gamma_l=100$, $\gamma_u=100$) with **$N_{min}=4$**. Notably, even with such extremely few tail class samples, our method still demonstrates significant performance gains over baselines.
>
> To further evaluate our method, we conducted additional experiments under **three more extreme settings**: **extreme sparsity setting A1** ($N_{max}=100$, $N_{min}=1$, $M_{max}=4900$, $\gamma_l=100$, $\gamma_u=100$), **extreme sparsity setting A2** ($N_{max}=10$, $N_{min}=1$, $M_{max}=4990$, $\gamma_l=10$,$\gamma_u=300), and **extreme imbalance setting B** ($N_{max}=400$, $N_{min}\approx1.33$ (1 in practice), $M_{max}=4600$, $\gamma_l=300$, $\gamma_u=300$).
> As demonstrated in Table R7, our method achieves state-of-the-art performance across three settings, attaining 69.56% accuracy in setting A1, 49.28% in setting A2, and 77.78% in setting B. These results represent significant improvements over all baseline methods, underscoring our method's exceptional capability to handle extreme settings.
> ##### Table R7
> |           | Extreme sparsity setting A1 | Extreme sparsity setting A2 | Extreme imbalance setting B |
> | --------- | ----- | ----- | ----- |
> | FreeMatch |63.86|45.75|55.92|
> | SimPro    |31.01|15.20|50.98|
> | Ours      |**69.56**|**49.28**|**77.78**|
> ### **Real-world datasets (Q3.1)**
> The Food-101 dataset (evaluated in Table 3) is a widely adopted benchmark for food image classification. Its training set naturally contains a proportion of label noise, making the data distribution implicitly unknown. Thus, distribution mismatches between labeled and unlabeled data occur when splitting the training set into labeled and unlabeled datasets. Despite these challenges, our method achieves a significant average accuracy gain of **2.57%** on this dataset.
>
> To further evaluate our method’s effectiveness in large-scale and real-world scenarios, we conduct experiments on ImageNet-127 (an imbalanced dataset, **127 classes**, **imbalance ratio 286**) with an arbitrary (non-long-tailed) distribution. As shown in Table R8, our method consistently outperforms baselines, achieving gains of **+1.27% at 32×32 resolution** and **+2.82% at 64×64 resolution**. These results highlight our method’s robustness in large-scale, realistic settings beyond standard benchmarks. As suggested, these results and corresponding analysis will be incorporated into **Section 4.2 Results** in the final version.
> ##### Table R8
> |           | 32 \*32 | 64 \*64 |
> | --------- | ----- | ----- |
> | FixMatch  |29.57|37.40|
> | FreeMatch |31.37|39.67|
> | SoftMatch |31.50|39.40|
> | ACR       |38.47|47.61|
> | SimPro    |43.31|46.93|
> | CDMAD     |15.46|21.92|
> | Ours      |**44.58**|**50.43**|
> ### **Real-world applications (Q3.2)**
> In medical applications, **distribution mismatch between labeled and unlabeled data occurs naturally**. Labeled clinical datasets from hospitals typically follow a long-tailed distribution, with abundant cases of common diseases (head classes) but very few cases of rare diseases (tail classes). In contrast, unlabeled data collected from broader populations typically includes a majority of healthy individuals and only a small proportion of diseased individuals, especially those with rare diseases. This creates a fundamental divergence between labeled and unlabeled data distributions.
>
> Furthermore, disease prevalence varies significantly across regions due to differences in climate, healthcare access, and other factors. Consequently, datasets collected from different geographical areas may exhibit arbitrary distribution patterns.
> ### **Evaluation using other architectures (Q4)**
> To evaluate architectural generalization, we conduct experiments using ResNet-50 on CIFAR-10-LT with ($N_{max}$=400, $M_{max}$=4600, $\gamma_l$=100, $\gamma_u$=100). The labeled data follows a long-tailed distribution, while the unlabeled data conforms to either an inverse long-tailed or an arbitrary distribution. As evidenced by Table R9, our approach demonstrates significant improvements, achieving absolute performance gains of **7.85%** and **7.23%** over current state-of-the-art methods in these respective scenarios. These results highlight our method's consistent robustness when applied to different network architectures and its ability to handle varying distribution patterns in the unlabeled data. The complete experimental results and detailed analysis will be incorporated into **Section 4.3 Analysis** of the final paper to further support these claims.
> ##### Table R9
> |           | Arbitrary | Inverse long-tailed |
> | --------- | ----- | ----- |
> | FreeMatch |47.68|48.28|
> | SimPro    |43.67|51.26|
> | Ours      |**55.53**|**58.49**|

---

> > ### Comment · Reviewer_HaKG · 2025-08-01
> >
> > Thank you for your detailed clarifications and additional experiments. I appreciate the effort to address each concern, and the new results under extreme settings are helpful.
> >
> > That said, I still find it difficult to raise my score. As currently framed, the proposed method appears to rely heavily on well-established components, including:
> >
> > Pseudo-label-driven distribution correction
> > Logit adjustment to achieve Bayes-optimal calibration
> > Consistency-based auxiliary learning
> > Class-aware enhancements for tail classes
> >
> > Each of these modules has clear precedents in recent literature (e.g., FlexDA, SimPro, UPS, DASO), and while the integration is well-executed, the conceptual novelty remains modest.
> >
> > Could the authors more explicitly clarify how their method differs from Flexible Distribution Alignment: Towards Long-tailed Semi-supervised Learning with Proper Calibration (ECCV 2024)? In particular:
> >
> > Both methods use pseudo-label distributions to guide re-balancing and calibration.
> > Both apply logit adjustment to mitigate class imbalance bias.
> > Both leverage low- and high-confidence unlabeled data through consistency-based regularization.
> >
> > Thank you again for the thoughtful and comprehensive rebuttal.

---

> > > ### Author Response · Authors · 2025-08-02
> > > **Further clarify the novelty of our method.**
> > >
> > > We sincerely appreciate the reviewer’s constructive feedback and are pleased to further clarify the novelty of our method and its distinctions from FlexDA.
> > >
> > > While FlexDA relies on estimating the unlabeled data distribution for distribution alignment and pseudo-label generation, our core contribution introduces a controllable self-reinforcing optimization cycle that **entirely bypasses unlabeled data distribution estimation and its novelty is confirmed by reviewers TLMt, ydDN, and Ckat**. This approach is fundamentally different from FlexDA’s dynamic distribution alignment. Specifically, we iteratively expand the labeled dataset with reliable pseudo-labels, train a Bayes-optimal classifier via logit adjustment on the updated labeled data **without using the pseudo-label distribution**. More importantly, we further theoretically prove that **this optimization cycle can significantly reduce the generalization error, whereas FlexDA lacks such theoretical guarantees**.
> > >
> > > Moreover, our auxiliary branch leverages all available samples (without a confidence threshold) to enhance feature learning while preventing the primary branch classifier from error pseudo-labels. Notably, prior methods (e.g., FlexDA, SimPro, UPS, DASO) lack class-aware enhancements for tail classes.
> > >
> > > Table R10 presents a comparative evaluation between our method and FlexDA (ECCV 2024) on CIFAR-10-LT with ($N_{max}=400$, $M_{max}=4600$, long-tailed labeled data distribution, arbitrary unlabeled data distribution). The results demonstrate that our approach outperforms FlexDA by a significant margin of **15-16%** in accuracy. This substantial performance gap further validates the superiority of our optimization cycle, which completely bypasses unlabeled data distribution estimation, as opposed to FlexDA's approach that relies on estimating the unlabeled distribution and achieving effective distribution alignment.
> > >
> > > These methodological advances, supported by theoretical guarantees and superior empirical results, underscore the conceptual and practical distinctions of our approach.
> > > ##### Table R10
> > > |           | $\gamma=100$ | $\gamma=150$ |
> > > | --------- | ----- | ----- |
> > > | FlexDA |65.58|62.39|
> > > | Ours      |**82.33**|**78.09**|

---

### Official Review · Reviewer_Ckat · 2025-06-30

**Clarity:** 3
**Significance:** 4
**Originality:** 4
**Rating:** 5
**Confidence:** 5

**Summary:**

This paper studies the long-tailed semi-supervised learning problem. Specifically, it investigates an under-explored and more realistic setting: the distribution of the unlabeled data is generally unknown and may follow an arbitrary distribution. This is a very challenging problem in semi-supervised learning. To solve this problem, this paper proposes a Controllable Pseudo-label Generation (CPG) framework, expanding the labeled dataset with the progressively identified reliable pseudo-labels from the unlabeled dataset and training the model on the updated labeled dataset with a known distribution, making it unaffected by the unlabeled data distribution. The proposed CPG contains a reliable pseudo-labels filtering mechanism and a Bayes-optimal classifier. This paper also proposes a class-aware adaptive augmentation module to enhance the representation of minority classes, as well as an auxiliary branch to maximize data utilization.

Generally, the idea is quite novel. Especially, the idea of handling the unknown label distribution of unlabeled samples is quite novel and reasonable. The proposed method is supported by a theory that demonstrates the proposed optimization cycle can significantly reduce the generalization error. The authors conducted extensive experiments to evaluate the proposed method, which suggests that it significantly outperforms previous methods by up to 16.29%. The visual illustration in this paper is informative (like Figure 2), and the writing of this paper is also well done. The code is available in the supplementary material.

**Questions:**

1. Different from the previous works in semi-supervised learning, this paper does not use the consistency between the strong augmentation and the weak augmentation. Please explain the reason.
2. The Bayes-optimal classifier can handle the tailed class samples well, so why is an additional class-aware adaptive augmentation adopted to enhance the minority class representations?
3. Why is an auxiliary branch adopted in the proposed method?
4. The proposed method does not estimate the distribution of the unlabeled samples. It just progressively incorporates more unlabeled samples in the labeled dataset. Will this progressive incorporation finally approximate the ground-truth label distribution of the unlabeled dataset?

**Ethical Concerns:**

["NO or VERY MINOR ethics concerns only"]

**Final Justification:**

This rebuttal have addressed all of my concerns. I keep my score.

**Limitations:**

Yes.

**Paper Formatting Concerns:**

None.

**Quality:**

4

**Strengths And Weaknesses:**

#### Strengths:

1. The investigated problem is interesting, under-explored, and more realistic.
2. The idea to handle the unknown unlabeled data is novel, and the adopted techniques are reasonable.
3. The authors theoretically prove that the proposed controllable self-reinforcing optimization cycle can significantly reduce the generalization error.
4. The experimental performance of the proposed method significantly outperforms the previous methods, and extensive ablation studies validate the effectiveness of each involved component in the proposed method.

#### Weakness:

1. Although the investigated problem is under-explored. I believe the techniques in long-tailed learning will help solve this problem. Therefore, this paper should review the related long-tailed learning methods in the "Related Work" section.
2. Please provide the pseudo-code in the paper, as it will help the authors to reproduce the results.

---

> ### Author Rebuttal · Authors · 2025-07-31
>
> ### **Review related long-tailed learning methods (W1)**
> We sincerely appreciate the reviewer’s insightful suggestion regarding the connection to long-tailed learning. In the revised manuscript, we will expand the **Related Work** section to include a dedicated discussion of long-tailed learning methods, particularly focusing on techniques that could inform solutions to our investigated problem.
> ### **Pseudo-code (W2)**
> For the reviewer's convenience, we would like to highlight that the pseudo-code of our proposed method was already included in Appnedix E of the original submission. We will move it in the main text in the final version.
> ### **Without consistency regularization (Q1)**
> While consistency regularization between augmentation views is commonly used in semi-supervised learning, we avoid it in our controllable self-reinforcing optimization cycle for two key reasons. First, pseudo-label quality is highly sensitive to the distribution mismatch between labeled and unlabeled data in long-tailed semi-supervised learning. Second, propagating error pseudo-labels from weak augmentation views to strong augmentation views can reinforce the model’s learning of incorrect predictions, leading to error accumulation and performance degradation (as demonstrated in Figures 1 and 3).
> ### **Why is an additional class-aware adaptive augmentation necessary (Q2)**
> While logit adjustment induced Bayes-optimal classifiers can handle long-tailed distribution, the fundamental challenges of limited sample size and low diversity in minority classes remain unavoidable in long-tailed learning. Our class-aware adaptive augmentation specifically targets these practical limitations by explicitly enhancing minority class representations.
> ### **Why use an auxiliary branch (Q3)**
> Our method progressively expands the labeled dataset by identifying reliable pseudo-labels from the unlabeled dataset. However, during early training stages when model performance is still weak, only a limited number of samples meet the reliability condition for pseudo-labeling (as demonstrated in Figures 1 and 3, Appendix Figure 5). To ensure full utilization of all available training samples throughout the learning process, we introduce an auxiliary branch that: (i) guarantees 100% sample participation from the beginning of training, and (ii) provides additional supervision signals to stabilize representation learning during the critical initial phase.
> ### **Will the progressive incorporation finally approximate the ground-truth label distribution of the unlabeled dataset (Q4)**
> While our method does not explicitly estimate the unlabeled data distribution, the progressive incorporation of reliable pseudo-labels through our controllable self-reinforcing optimization cycle naturally approximates the ground-truth unlabeled data distribution as training progresses. This is because the model’s increasing confidence enables it to gradually include more diverse samples, including those from minority classes, improving distribution alignment. **Empirical results in Appendix Figure 5 demonstrate that the pseudo-label distribution converges toward the ground-truth unlabeled data distribution over time**.

---

> > ### Author Response · Authors · 2025-08-04
> >
> > Dear Reviewer **Ckat**,
> >
> > Thanks again for your time and effort in reviewing this paper. We appreciate your positive recommendation. If you have any other suggestions or comments, we would be more than happy to give more explanations. Thanks.
> >
> > Regards from the authors.

---

> > > ### Comment · Reviewer_Ckat · 2025-08-05
> > >
> > > Thank you for your thoughtful responses. Your clarifications have addressed all of my concerns. I keep my score.

---

> ### Author Response · Authors · 2025-08-05
>
> Dear Reviewer **Ckat**,
>
> Thanks again for your valuable comments. We are glad that **our clarifications have addressed all of your concerns**. Please raise the rating as your opinion is important to this work.
>
> Regards from the authors.

---

### Official Review · Reviewer_ydDN · 2025-07-02

**Clarity:** 3
**Significance:** 2
**Originality:** 3
**Rating:** 4
**Confidence:** 4

**Summary:**

This paper addresses the challenge of semi-supervised learning when the distribution of unlabeled data is arbitrary and unknown.
The authors propose the CPG framework, which selectively adds only reliable pseudo-labels to the labeled set from unlabeled set, and applies logit adjustment based on the known class distribution.
They further introduce class-aware adaptive augmentation and an auxiliary branch to improve minority class representation and make full use of both labeled and unlabeled data.
Experiments on several long-tailed benchmarks, including CIFAR-10-LT, CIFAR-100-LT, Food-101-LT, and SVHN-LT, show that CPG significantly outperforms previous methods, with gains of up to 16 percentage points.
Theoretical analysis also supports the approach by showing that the iterative pseudo-labeling process helps reduce generalization error.

**Questions:**

- Since equation (3) uses a confidence threshold to filter pseudo-labels, it would be helpful to provide an ablation study or sensitivity analysis showing how different threshold values affect the overall performance,
- As the method assumes noise-free labeled data, it would improve the paper to include experiments evaluating the robustness of CPG under various levels of label noise, as well as comparisons with baseline methods such as FreeMatch and SimPro.
- It would further strengthen the paper if the effectiveness of the proposed method could be proved on large-scale or real-world datasets beyond standard benchmarks.

**Ethical Concerns:**

["NO or VERY MINOR ethics concerns only"]

**Final Justification:**

The authors have addressed my main concerns in the rebuttal, providing additional experiments that demonstrate the method can learn tail classes effectively, and that it is robust to different confidence thresholds, label noise, and real-world large-scale datasets.
Given these clarifications and the strong empirical results, I maintain my rating as borderline accept.

**Limitations:**

yes

**Quality:**

3

**Strengths And Weaknesses:**

Strengths:
- While previous methods require prior assumptions about the distribution of unlabeled data, CPG can operate robustly even under arbitrary unlabeled data distributions. Moreover, unlike previous ReaLTSSL approaches (e.g., SimPro) that estimate the unlabeled data distribution, CPG only accumulates reliable samples as labeled data, resulting in a more robust training process.
- The proposed key components (CPG, CAA, auxiliary branch) are thoroughly ablated, and the paper provides detailed analysis across various distribution settings
- The paper is easy to follow.
- The proposed method is theoretically justified with formal proofs.
- The authors provide code to ensure the reproducibility of their results.

Weaknesses:
- The CPG framework accumulates only reliable pseudo-labels from the unlabeled data into the labeled pool, which poses a risk that tail class samples may be entirely omitted in certain situations. In such cases, the corresponding classes are completely excluded from training and loss balancing (logic-adjustment), potentially resulting in the model failing to learn or classify tail classes at all.
- Equation (3) uses a confidence threshold to filter reliable pseudo-labels. However, since the performance of the proposed method is likely to be highly sensitive to the choice of this threshold, the paper lacks sufficient ablation analysis regarding this hyper parameter.
- As mentioned in Section H (Limitation), the current method may encounter problems when noisy labels are present. In real-world scenarios, label noise can exist and potentially harm performance. Therefore, it would be valuable to report experimental results analyzing how CPG performs compared to baseline methods (e.g., FreeMatch, SimPro) under varying levels of label noise in the dataset.

---

> ### Author Rebuttal · Authors · 2025-07-31
>
> ### **Potentially fail to learn tail classes (W1)**
> We sincerely appreciate the reviewer’s insightful comments. We would like to clarify that our model does not fail to learn or classify tail classes. While the labeled data follows a long-tailed distribution, **it does contain some tail class samples in the labeled dataset**. Our approach enhances tail classes through logit adjustment and the proposed class-aware adaptive augmentation module, ensuring robust representation even when initial pseudo-label selection misses tail class samples. As shown in Figure 1, our method can generate more reliable pseudo-labels than FreeMatch and SimPro in tail classes like classes 1 and 2. Moreover, as demonstrated in Appendix Figure 5, our method progressively identifies nearly all samples as reliable pseudo-labels during training, with their final distribution closely approximating the true unlabeled data distribution.
> ### **Different confidence threshold selection (W2 & Q1)**
> The confidence threshold $t=0.95$ is a common setting in semi-supervised learning, which **we adopt for fair comparison**. To evaluate the impact of different confidence thresholds on model performance, we conduct additional experiments on CIFAR-10-LT (using the same experimental setup as in Table R1) with thresholds of $t=0.85$ and $t=0.75$. As shown in Table R4, our method’s performance variations across different thresholds remain relatively small (within 1-2%), and it maintains consistently superior performance compared to baseline approaches. This threshold-insensitive property highlights the stability of our method, and these results will be incorporated into the final version of the paper in **Section 4.3 Analysis**.
> ##### Table R4
> | Unlabeled distribution | Arbitrary | Inverse long-tailed | Long-tailed |
> | --- | --- | --- | --- |
> | FreeMatch |66.33|68.10|69.42|
> | SimPro   |64.73|65.25|64.37|
> | Ours ($t=0.75$) |81.35|78.26|76.20|
> | Ours ($t=0.85$) |81.89|80.85|77.79|
> | Ours ($t=0.95$)|**82.33**|**82.32**|**78.35**|
> ### **Noisy scenarios (W3 & Q2)**
> To comprehensively evaluate our method’s robustness under noisy scenarios, we conduct additional experiments on CIFAR-10-LT ($N_{max}$=400, $M_{max}$=4600, $\gamma_l$=100, $\gamma_u$=100) under varying noise rates (0%, 10%, 20%, and 30%). The labeled data follows a long-tailed distribution, while the unlabeled data adheres to an arbitrary distribution. As demonstrated in Table R5, our method consistently outperforms baselines in noisy settings, achieving the highest accuracy across all noise rates with minimal performance degradation. Notably, when noise increases from 0% to 30%, our method exhibits a degradation (Δ) of only **−12.46%**, compared to **−15.04%** for FreeMatch and **−29.93%** for SimPro. This demonstrates our method's strong robustness to label noise, and these results will be added to **Section 4.3 Analysis** in the final paper.
> ##### Table R5
> | Noise rate | 0% | 10% | 20% | 30% | Δ(0→30%) |
> | --- | --- | --- | --- | --- | --- |
> | FreeMatch |66.33|60.33|56.24|51.29|-15.04|
> | SimPro    |64.73|59.86|44.54|34.80|-29.93|
> | Ours      |**82.33**|**75.95**|**71.82**|**69.87**|**-12.46**|
> ### **Large-scale or real-world datasets (Q3)**
> To further evaluate our method’s effectiveness in large-scale and real-world scenarios, we conduct experiments on ImageNet-127 (an imbalanced dataset, **127 classes**, **imbalance ratio 286**) with an arbitrary (non-long-tailed) distribution. As shown in Table R6, our method consistently outperforms baselines, achieving gains of **+1.27% at 32×32 resolution** and **+2.82% at 64×64 resolution**. These results highlight our method’s robustness in large-scale, realistic settings beyond standard benchmarks. As suggested, these results and corresponding analysis will be incorporated into **Section 4.2 Results** in the final version.
> ##### Table R6
> |           | 32 \*32 | 64 \*64 |
> | --------- | ----- | ----- |
> | FixMatch  |29.57|37.40|
> | FreeMatch |31.37|39.67|
> | SoftMatch |31.50|39.40|
> | ACR       |38.47|47.61|
> | SimPro    |43.31|46.93|
> | CDMAD     |15.46|21.92|
> | Ours      |**44.58**|**50.43**|

---

> > ### Author Response · Authors · 2025-08-05
> >
> > Dear Reviewer **ydDN**,
> >
> > We appreciate your positive recommendation. In the rebuttal, we clarify that **our model can learn tail classes effectively** (as shown in Figure 1 of the paper, our model can learn tail classes like classes 1 and 2 better than baselines; Appendix Figure 5 of the paper demonstrates our method can progressively identify nearly all samples as reliable pseudo-labels during training). Furthermore, in the rebuttal, we expand our evaluation in the following aspects:
> >
> > 1.We conduct additional experiments on CIFAR-10-LT with thresholds of $t=0.85$ and $t=0.75$, and the results in Table R4 demonstrate **our method is threshold-insensitive**.
> >
> > 2.We conduct additional experiments under noisy scenarios, and the results in Table R5 demonstrate that **our method is more robust to noisy labels than baselines**.
> >
> > 3.We conduct additional experiments on large-scale and real-world datasets, and the results in Table R6 **highlight our method’s robustness in large-scale, realistic settings beyond standard benchmarks**.
> >
> > Additionally, the following Table R11 presents the head, medium, and tail classes accuracy on **CIFAR-10-LT ($N_{max}=400$, $M_{max}=4600$, $\gamma_l=100$, $\gamma_u=100$)**, the labeled data follows a long-tailed distribution, while the unlabeled data varies across arbitrary and long-tailed distributions. Notably, our method achieves 89.58% accuracy on tail classes under arbitrary unlabeled data distribution (vs. 53.60% for FreeMatch and 49.55% for SimPro) and 72.90% accuracy under long-tailed unlabeled data distribution (vs. 52.43% for FreeMatch and 34.25% for SimPro), **such substantial accuracy improvements in tail classes demonstrate that our method can learn tail classes effectively**.
> >
> > ##### Table R11
> > | Arbitrary | Head | Medium | Tail |
> > | --- | --- | --- | --- |
> > | FreeMatch |85.63|64.00|53.60|
> > | SimPro |84.00|65.70|49.55|
> > | Ours |84.27|**70.73**|**89.58**|
> > | Long-tailed | Head | Medium | Tail |
> > | FreeMatch |90.13|71.37|52.43|
> > | SimPro |95.10|73.80|34.25|
> > | Ours |89.67|**74.30**|**72.90**|
> >
> > If there are any other questions or points you would like us to clarify further, we are more than willing to assist. Thank you again for your contribution to the review process.
> >
> > Regards from the authors.

---

> ### Author Response · Authors · 2025-08-06
>
> Reviewer **ydDN**,
>
> We appreciate your valuable comments, which are important to improve the quality of this paper. We also appreciate your positive rating. **As the deadline for the author-reviewer discussion period is approaching**, please check whether your previous concerns have been fully addressed.
>
> Looking forward to your reply. Thanks.
>
> Regards.

---

> > ### Author Response · Authors · 2025-08-07
> >
> > Reviewer **ydDN**,
> >
> > Thanks again for your valuable comments on this paper. We are glad that your preliminary rating of this paper is positive, which is very encouraging to us. You have posed some questions in the comments. We have tried our best to address them. Please kindly check whether your previous concerns have been fully addressed. Thanks.
> >
> > Looking forward to your reply.
> >
> > Regards.

---

> > > ### Comment · Reviewer_ydDN · 2025-08-08
> > >
> > > The authors have addressed all of my concerns. Moreover, I was able to confirm that the proposed method performs well even in noisy and real-world settings. I appreciate the effort put into the rebuttal and will maintain my score as borderline accept.

---

> > > > ### Author Response · Authors · 2025-08-08
> > > >
> > > > Dear Reviewer **ydDN**,
> > > >
> > > > Thanks for your recognition of our work. We are glad that all your concerns have been addressed. We will follow your suggestions to revise the paper in the final version. Thanks again for your time and effort in reviewing this paper.
> > > >
> > > > Regards from the authors.

---

### Official Review · Reviewer_TLMt · 2025-07-03

**Clarity:** 3
**Significance:** 3
**Originality:** 3
**Rating:** 5
**Confidence:** 4

**Summary:**

The paper studies the problem of Pseudo-label generation under a Realistic Long-Tailed Semi-Supervised Learning (ReaLTSSL) setting. In this setting, a long-tailed distribution (characterized by an imbalance ratio) is combined with an unlabeled set that has unknown and even arbitrary distribution.

The authors proposed a iterative method to generate labels for the unlabeled set. The method keeps the pseudo-label for unlabeled images only when its weakly-augmented and strongly-augmented views predict the same class and both predictions exceed a preset confidence threshold. Then the accepted images are moved into the labeled set and the classifier is retrained with logit adjustment. This process is then repeated for more labels.

The paper also proposed two add-ons CAA (help with minority classes) and an auxiliary FlixMatch-style branch for consistency regularization.

The method is then evaluated on 4 commonly used benchmarks and achieves a 16.29% performance gain on CIFAR-10-LT outperforming previous methods for long-tailed SSL. The authors also provided theoretical analysis on the optimization cycle.

**Questions:**

1. We see a huge performance gain for CIFAR-10-LT, but the performance gain is less significant for CIFAR-100-LT. Is it possible to provide some reasoning on why the performance gain is less significant for a similar dataset with higher class cardinality?

2. The paper uses confidence interval $\tau=0.95$ to filter out the pseudo-labels. Is it possible to use a different confidence interval, and how does it affect the performance?

3. The experiments are conducted on long-tailed datasets with class imbalance. Could you also give macro-F1 for the main CIFAR-10-LT and CIFAR-100-LT results? A quick per-class breakdown or confusion matrix would make it clear where CPG’s gains come from.

4. Have you tried CPG on non-vision data (e.g., text, audio)? What components would need to change for other modalities?

5. In line 470, for risk of both the labeled and unlabeled dataset, the data $x_i$ ranges starting from $1$. Maybe it is possible to use notation to differentiate the two or let the latter one start from $N+1$?

**Ethical Concerns:**

["NO or VERY MINOR ethics concerns only"]

**Final Justification:**

The authors conducted experiments with Macro-F1, which further strengthens the claim that the proposed method works well with class imbalance. The part that CIFAR-100-LT is inherently harder and thus the performance gain is smaller convinced me, and the new audio-modality experiments broaden the method’s demonstrated scope. I also appreciate the clearer discussion of how CPG diverges from related approaches, as well as the improved notation in the final paper. As a result, I maintain my recommendation for acceptance.

**Limitations:**

Yes

**Paper Formatting Concerns:**

No Paper Formatting Concerns.

**Quality:**

4

**Strengths And Weaknesses:**

Quality:
Strength: The paper did rigorous evaluation on four datasets (CIFAR-10-LT/100-LT, Food-101-LT, SVHN-LT) across multiple imbalance ratios. The paper also conducted ablation studies to isolate the contribution of each module. The paper provided a well-documented code in the supplementary material that is clear and easy to follow. The paper provided a theoretical risk-reduction bound (Theorem 1) and its practical implications are discussed (lines 249-260).
Weakness: The proposed method CPG is supposed to target the long-tailed distribution with class imbalance. However, only accuracy is reported; macro-F1 or other metrics are not reported.

Clarity:
Strength: The paper is well-written and easy to follow. The algorithm is concisely conveyed in Algorithm 1 together with previously defined equations.
Weakness: The plots on page 9 feel crowded and break the reading flow.

Significance:
Strength: CPG achieves up to 16.29% performance gain on CIFAR-10-LT. The advantage of CPG is clear in Figure 1.
Weakness: Performance gain on CIFAR-100-LT is lower compared with CIFAR-10-LT. A discussion of why performance drops as class cardinality grows would strengthen the claim of broad significance. The paper only investigated the performance of computer vision datasets. It is not clear how the proposed method performs on other modalities.

Originality:
Weakness: The starting filter process uses weak/strong-view confidence. This is similar to FixMatch/FlexMatch. The paper can highlight more clearly how CPG diverges from the other methods conceptually in the appendix.
Strength: CPG only keeps an unlabeled sample when weak and strong views agree (stricter than Fix/FlexMatch). The accepted samples are promoted to the labeled pool, and then a logit-adjusted head is retrained with pseudo-labels and logit adjustment, which helps with class imbalance.

---

> ### Author Rebuttal · Authors · 2025-07-31
>
> ### **The proposed method is supposed to target the long-tailed distribution. However, only accuracy is reported (W1 & Q3)**
> We appreciate the reviewer’s insightful comments. To better evaluate our method‘s performance under class imbalance, we have now included the Macro-F1 metric in our analysis. Table R1 presents results on two benchmarks: (i) **CIFAR-10-LT ($N_{max}=400$, $M_{max}=4600$, $\gamma_l=100$, $\gamma_u=100$)** in the left four columns, and (ii) **CIFAR-100-LT ($N_{max}=50$, $M_{max}=450$, $\gamma_l=10$, $\gamma_u=10$)** in the right four columns. In both cases, the labeled data follows a long-tailed distribution, while the unlabeled data varies across long-tailed, inverse long-tailed, and arbitrary distributions.
>
> The results in Table R1 show that our method achieves statistically significant improvements in Macro-F1 over existing baselines, further validating its effectiveness for imbalance learning. Notably, it yields average Macro-F1 gains of **13.84%** on CIFAR-10-LT and **4.55%** on CIFAR-100-LT. These results and analysis will be added to **Section 4.2 Results** in the final version.
> ##### Table R1
> | Unlabeled distribution | Arbitrary | Inverse long-tailed | Long-tailed | Avg. | Arbitrary | Inverse long-tailed | Long-tailed | Avg. |
> | --- | --- | --- | --- | --- | --- | --- | --- | --- |
> | FixMatch |53.52|54.24|66.47|58.07|44.58|44.85|46.46|45.30|
> | FreeMatch|66.00|67.00|68.04|67.01|44.13|43.57|43.84|43.85|
> | SoftMatch|62.49|66.66|71.53|66.89|45.67|46.73|46.91|46.44|
> | ACR      |59.01|64.03|70.25|64.43|45.69|48.89|41.92|45.50|
> | SimPro   |61.22|62.11|57.81|60.38|39.98|41.54|41.87|41.13|
> | CDMAD    |61.73|62.88|62.11|62.24|32.97|35.80|36.68|35.15|
> | Ours     |**81.99**|**82.31**|**78.25**|**80.85**|**51.31**|**51.25**|**50.41**|**50.99**|
> ### **The plots on page 9 feel crowded and break the reading flow (W2)**
> As suggested, we will improve the clarity and layout of the plots on page 9 to enhance readability in the final version.
> ### **Performance drops as class cardinality grows (for example on CIFAR-100-LT) (W3.1 & Q1)**
> We respond to the reviewer’s concerns regarding the relationship between performance gains and class cardinality.
> CIFAR-100 and CIFAR-10 have the same total training set size, but CIFAR-100 divides the data into 100 fine-grained classes (compared to 10 in CIFAR-10), reducing per-class samples by an order of magnitude. The finer class granularity also increases inter-class similarity, making classification inherently more challenging. As a result, the upper-bound performance (e.g., supervised learning) on CIFAR-100-LT (**64.62%**) is significantly lower than on CIFAR-10-LT (**85.22%**).
>
> Moreover, consistency regularization enhances model stability by encouraging similar predictions for different augmentations of the same sample. This promotes robust representations, preventing severe performance degradation and ensuring that the lower-bound performance on CIFAR-100-LT remains relatively stable.
>
> Given the narrower performance gap between upper and lower bounds in CIFAR-100-LT (compared to CIFAR-10-LT), the absolute improvement achievable by our method is inherently constrained. Nevertheless, our method still achieves substantial average accuracy gains of **3.09%** on CIFAR-100-LT (as shown in Table 2). These analysis will be added to **Section 4.2 Results** in the final version.
> ### **Evaluation on other modalities (W3.2 & Q4)**
> While our method is evaluated on vision tasks, **the core principles of CPG can be extended to other modalities**. In this rebuttal, we further evaluate our method on the **audio modality**. Specifically, for audio data, this adaptation requires two key modifications: (i) replacing vision-specific backbones (e.g., WideResNet-28-2) with audio-compatible models like HuBERT, and (ii) substituting image augmentations with audio-specific ones. To demonstrate this adaptability, we perform our CPG on ESC-50, a standard benchmark for environmental sound classification. As demonstrated in Table R2, our method achieves state-of-the-art performance on this audio benchmark, confirming its effectiveness beyond visual modality. These results strongly support the generalizability of CPG's framework across different modalities. These results and analysis will be added to **Section 4.2 Results** in the final version.
> ##### Table R2
> | FreeMatch | SimPro | Ours |
> | --- | --- | --- |
> | 68.13 | 68.75 | **69.13** |
> ### **Clarify how CPG diverges from other methods like FixMatch/FlexMatch (W4)**
> While CPG adopts confidence-based filtering similar to FixMatch/FlexMatch, its key innovation is the **controllable self-reinforcing optimization cycle**, which incorporates reliable pseudo-labels from the unlabeled dataset into the labeled dataset rather than employing consistency regularization, distinguishing it from existing methods. This cycle operates in three steps: (i) expanding the labeled dataset with reliable pseudo-labels, (ii) constructing a Bayesian classifier, and (iii) iteratively improving pseudo-label quality, forming a theoretically grounded feedback loop. Additionally, CPG ensures full sample utilization via an auxiliary branch and reduces label noise accumulation through a voting-based stabilization technique, which enforces consistent pseudo-label assignments across training steps. As evidenced by the ablation studies in Table 5, these components individually contribute significant performance improvements, with the auxiliary branch and optimization cycle yielding average gains of **1.82%** and **6.97%**, respectively.
> ### **Different confidence threshold selection (Q2)**
> The confidence threshold $t=0.95$ is a common setting in semi-supervised learning, which **we adopt for fair comparison**. To evaluate the impact of different confidence thresholds on model performance, we conduct additional experiments on CIFAR-10-LT (using the same experimental setup as in Table R1) with thresholds of $t=0.85$ and $t=0.75$. As shown in Table R3, our method’s performance variations across different thresholds remain relatively small (within 1-2%), and it maintains consistently superior performance compared to baseline approaches. This threshold-insensitive property highlights the stability of our method, and these results will be incorporated into the final version of the paper in **Section 4.3 Analysis**.
> ##### Table R3
> | Unlabeled distribution | Arbitrary | Inverse long-tailed | Long-tailed |
> | --- | --- | --- | --- |
> | FreeMatch |66.33|68.10|69.42|
> | SimPro   |64.73|65.25|64.37|
> | Ours ($t=0.75$) |81.35|78.26|76.20|
> | Ours ($t=0.85$) |81.89|80.85|77.79|
> | Ours ($t=0.95$)|**82.33**|**82.32**|**78.35**|
> ### **Clarify equation notation (Q5)**
> To improve notational clarity, we will modify the indexing in line 470 so that labeled data indices run from 1 to $ N $ and unlabeled data indices run from $N+1$ to $N+M$ in the final version.

---

> > ### Comment · Reviewer_TLMt · 2025-08-06
> >
> > Thank you for conducted the new experiments with Macro-F1, which further strengthens the claim that the proposed method works well with class imbalance. The part that CIFAR-100-LT is inherently harder and thus the performance gain is smaller convinced me, and the new audio-modality experiments broaden the method’s demonstrated scope. I also appreciate the clearer discussion of how CPG diverges from related approaches, as well as the improved notation in the final paper. The answers addressed my comments, and I will maintain my recommendation for acceptance. Hope to see the new experiments and results incorporated in the final paper.

---

> ### Author Response · Authors · 2025-08-06
>
> Dear Reviewer **TLMt**,
>
> We are glad your concerns have all been solved. We appreciate your recommendation for acceptance.  We will include the new experiments and results in the final paper. Thanks.
>
> Regards.

---

### Note · Authors · 2025-08-13

We thank all reviewers (TLMt, ydDN, Ckat, HaKG) for their constructive feedback.

Overall, the reviewers acknowledged the following strengths of our work:

1. Addressing the under-explored realistic long-tailed semi-supervised learning with unknown and arbitrary unlabeled data distributions. (All reviewers)

2. The controllable self-reinforcing optimization cycle innovatively bypasses unreliable distribution estimation, and its novelty is confirmed (All reviewers) and supported by the theoretical risk-reduction bound (TLMt, ydDN, Ckat).

4. Introducing class-aware adaptive augmentation and an auxiliary branch to improve minority class representation and make full use of all available data, respectively. (All reviewers)

5. Achieving SOTA results across multiple benchmarks by a large margin. (All reviewers)

6. Well-written paper, and publicly released code for easy reproducibility and following. (TLMt, ydDN, Ckat)

7. Clear ablation studies validate the contribution of each component. (All reviewers)

8. The design of diverse experimental settings strengthens the generality of the results. (HaKG)

9. The use of pairwise t-tests provides strong statistical support for performance claims. (HaKG)

Meanwhile, following the reviewers’ suggestions, our main responses are as follows:

1. We have included the Macro-F1 metric in our analysis to better evaluate performance under class imbalance (Table R1, TLMt).

2. We have clarified the paper's content, including plot layouts and notation (TLMt), our model does not fail on tail classes (Table R11, ydDN), the pseudo-code (Ckat), progressive incorporation finally approximating true unlabeled data distribution (Ckat), and the real-world applications (HaKG).

3. Added results on an audio dataset (Table R2, TLMt) and a large-scale real-world dataset (Table R6/8, ydDN, HaKG) prove the robustness and generalizability of our method.

4. Clearly differentiating CPG from related methods like FixMatch/FlexMatch (TLMt) and UPS/CADR/FlexDA (HaKG), highlighting its main novelty (e.g., controllable self-reinforcing optimization cycle).

5. We conducted ablation studies to demonstrate that our method is stable across different confidence thresholds (Table R3/4, TLMt, ydDN) and robust to varying levels of label noise (Table R5, ydDN), extreme settings (Table R7, HaKG), and backbone architectures (Table R9, HaKG).

We believe these additional experiments, clarifications, and textual improvements fully address all reviewer concerns.

---

### Decision · Program_Chairs · 2025-09-17

**Decision:**

Accept (poster)

**Comment:**

The manuscript addresses the problem of long-tailed semi-supervised learning. The main contributions are a method for controllable pseudo-label generation, its theoretical analysis, a class-aware augmentation, and the experimental evaluation. While the theoretical analysis and the evaluation are the main strengths of the generally well-written manuscript, the novelty of the approach is limited compared to FixMatch/FlexMatch/UPS/CADR/FlexDA. However, on a detail level, the authors argue convincingly in relation to those methods and the results together with the theoretical analysis justify acceptance of the paper.
After rebuttal and discussion three reviewers are rating accept (2) or BA, only HaKG suggest to reject (BR). Most of the pros and cons above a shared among the reviewers, including the criticism regarding novelty. HaKG argues that in particular FlexDA is very similar, but also points out that acceptance would be ok. Thus the AC reads this as a plain borderline and suggests to accept the manuscript.